# Targeting the fatty acid binding proteins disrupts multiple myeloma cell cycle progression and MYC signaling

**Mariah Farrell**[1,2,3], **Heather Fairfield**[1,2,3], **Michelle Karam**[1], **Anastasia D'Amico**[1], **Connor S Murphy**[1,2], **Carolyne Falank**[1], **Romanos Sklavenitis Pistofidi**[4,5], **Amanda Cao**[4,5], **Catherine R Marinac**[4,5], **Julie A Dragon**[6], **Lauren McGuinness**[1,7], **Carlos G Gartner**[1,2,3], **Reagan Di Iorio**[1,7], **Edward Jachimowicz**[1], **Victoria DeMambro**[1,2], **Calvin Vary**[1,2,3], **Michaela R Reagan**[1,2,3]*

[1]Center for Molecular Medicine, Maine Health Institute for Research, Scarborough, United States; [2]Graduate School of Biomedical Science and Engineering, University of Maine, Orono, United States; [3]Tufts University School of Medicine, Boston, United States; [4]Dana-Farber Cancer Institute, Boston, United States; [5]Harvard Medical School, Boston, United States; [6]University of Vermont, Burlington, United States; [7]University of New England, Biddeford, United States

**Abstract** Multiple myeloma is an incurable plasma cell malignancy with only a 53% 5-year survival rate. There is a critical need to find new multiple myeloma vulnerabilities and therapeutic avenues. Herein, we identified and explored a novel multiple myeloma target: the fatty acid binding protein (FABP) family. In our work, myeloma cells were treated with FABP inhibitors (BMS3094013 and SBFI-26) and examined in vivo and in vitro for cell cycle state, proliferation, apoptosis, mitochondrial membrane potential, cellular metabolism (oxygen consumption rates and fatty acid oxidation), and DNA methylation properties. Myeloma cell responses to BMS309403, SBFI-26, or both, were also assessed with RNA sequencing (RNA-Seq) and proteomic analysis, and confirmed with western blotting and qRT-PCR. Myeloma cell dependency on FABPs was assessed using the Cancer Dependency Map (DepMap). Finally, MM patient datasets (CoMMpass and GEO) were mined for *FABP* expression correlations with clinical outcomes. We found that myeloma cells treated with FABPi or with *FABP5* knockout (generated via CRISPR/Cas9 editing) exhibited diminished proliferation, increased apoptosis, and metabolic changes in vitro. FABPi had mixed results in vivo, in two pre-clinical MM mouse models, suggesting optimization of in vivo delivery, dosing, or type of FABP inhibitors will be needed before clinical applicability. FABPi negatively impacted mitochondrial respiration and reduced expression of MYC and other key signaling pathways in MM cells in vitro. Clinical data demonstrated worse overall and progression-free survival in patients with high *FABP5* expression in tumor cells. Overall, this study establishes the FABP family as a potentially new target in multiple myeloma. In MM cells, FABPs have a multitude of actions and cellular roles that result in the support of myeloma progression. Further research into the FABP family in MM is warrented, especially into the effective translation of targeting these in vivo.

## Editor's evaluation

This paper has valuable findings that have practical implications within multiple myeloma and tumor microenvironment fields. It describes a family of genes regulating myeloma cell survival and proliferation. The approaches used are convincing, state-of-the-art, and rigorous. The data support the essential claims of the manuscript and have mechanistic depth.

*For correspondence:
Michaela.Reagan@MaineHealth.org

**eLife digest** Multiple myeloma is a type of blood cancer for which only a few treatments are available. Currently, only about half the patients with multiple myeloma survive for five years after diagnosis. Because obesity is a risk factor for multiple myeloma, researchers have been studying how fat cells or fatty acids affect multiple myeloma tumor cells to identify new treatment targets.

Fatty acid binding proteins (FABPs) are one promising target. The FABPs shuttle fatty acids and help cells communicate. Previous studies linked FABPs to some types of cancer, including another blood cancer called leukemia, and cancers of the prostate and breast. A recent study showed that patients with multiple myeloma, who have high levels of FABP5 in their tumors, have worse outcomes than patients with lower levels. But, so far, no one has studied the effects of inhibiting FABPs in multiple myeloma tumor cells or animals with multiple myeloma.

Farrell et al. show that blocking or eliminating FABPs kills myeloma tumor cells and slows their growth in a dish (in vitro) and in some laboratory mice. In the experiments, the researchers treated myeloma cells with drugs that inhibit FABPs or genetically engineered myeloma cells to lack FABPs. They also show that blocking FABPs reduces the activity of a protein called MYC, which promotes tumor cell survival in many types of cancer. It also changed the metabolism of the tumor cell. Finally, the team examined data collected from several sets of patients with multiple myeloma and found that patients with high FABP levels have more aggressive cancer.

The experiments lay the groundwork for more studies to determine if drugs or other therapies targeting FABPs could treat multiple myeloma. More research is needed to determine why inhibiting FABPs worked in some mice with multiple myeloma but not others, and whether FABP inhibitors might work better if combined with other cancer therapies. There were no signs that the drugs were toxic in mice, but more studies must prove they are safe and effective before testing the drugs in humans with multiple myeloma. Designing better or more potent FABP-blocking drugs may also lead to better animal study results.

## Introduction

Fatty acid binding protein (FABP) family members are small (12–15 kDa) proteins that reversibly bind lipids (*Hotamisligil and Bernlohr, 2015*). The 10 human FABP isoforms are functionally and spatially diverse, consisting of ten anti-parallel beta sheets, which form a beta barrel that shuttles fatty acids across membranes of organelles including peroxisomes, mitochondria, nuclei, and the endoplasmic reticulum (*Furuhashi and Hotamisligil, 2008*). FABPs influence cell structure, intracellular and extracellular signaling, metabolic and inflammatory pathways (*Hotamisligil and Bernlohr, 2015*), and maintain mitochondrial function (*Field et al., 2020*). While most cell types express a single FABP isoform, some co-express multiple FABPs that can functionally compensate for each other if needed (*Hotamisligil et al., 1996*; *Shaughnessy et al., 2000*), suggesting that broad FABP targeting may be necessary. FABP insufficiencies in humans and mice induce health benefits (eg. protection from cardiovascular disease, atherosclerosis, and obesity-induced type 2 diabetes), suggesting these to be safe therapeutic targets (*Cao et al., 2006*; *Maeda et al., 2005*; *Tuncman et al., 2006*).

Multiple myeloma (MM), a clonal expansion of malignant plasma cells, accounts for ~10% of hematological neoplasms (*Rajkumar, 2020*). Myeloma cell growth initiates in and spreads throughout the bone marrow, leading to aberrant cell proliferation and destruction of the bone (*Fairfield et al., 2016*). Treatments for myeloma patients have greatly improved within the past two decades (*American cancer institute, 2022*), but most patients eventually relapse, demonstrating the need to pursue more novel types of MM treatment. Few therapies are designed to specifically target molecules involved in the MM cell metabolism, despite recent findings that MM cells uptake fatty acids through fatty acid transport proteins, which can enhance their proliferation (*Panaroni et al., 2022*). Links between FABP4 and cancer have been demonstrated in prostate, breast, and ovarian cancer, and acute myeloid leukemia (AML; *Al-Jameel et al., 2017*; *Carbonetti et al., 2019*; *Herroon et al., 2013*; *Lan et al., 2011*; *Mukherjee et al., 2020*; *Shafat et al., 2017*; *Yan et al., 2018*; *Zhou et al., 2019*). FABP5 has been less widely studied in cancer, but is known to transport ligands to PPARD (*Tan et al., 2002*), which can intersect with many pro-tumor pathways that increase proliferation, survival (*Adhikary et al., 2013*; *Di-Poï et al., 2002*; *Tan et al., 2001*), and angiogenesis (*Wang et al., 2006*),

and decrease tumor suppressor expression (*Tan et al., 2001*). Herein we explored the oncogenic function of the FABPs in MM by examining therapeutic targeting with FABP inhibitors (FABPi) in multiple cell lines in vitro, and using genetic knockout of *FABP5*, pre-clinical models, large cell line datasets, and multiple patient datasets. Our results suggest FABPs are a novel target in MM due to the plethora of important biological functions that FABPs modulate to control cellular processes at multiple levels.

## Results

### *FABP5* is vital for MM cells and genetic knockout results in reduced cell number

We first examined FABP gene expression in MM cell lines and found that *FABP5* was the most highly-expressed FABP isoform in GFP+/Luc+MM.1S and RPMI-8226 cells (*Supplementary file 1*, *Fairfield et al., 2021*) and that some other FABPs were also expressed to a lesser extent (eg. *FABP3*, *FABP4*, and *FABP6*). FABP5 protein was also robustly expressed in these cells (*Figure 1A*, *Figure 1—figure supplement 1A*), and FABP5 consistently showed the expression in haematopoetic/lymphoid lineage lines within the Cancer Cell Line Encyclopedia (CCLE) at the gene level (*Figure 1—figure supplement 1B*) and protein level (*Figure 1—figure supplement 1C*, "*DepMap 22Q2, 2022*; *Ghandi et al., 2019*; *Nusinow et al., 2020*). In MM cell lines specifically, *FABP5* was the most highly expressed at the gene level (*Figure 1B*) and FABP5 and FABP6 were the most highly expressed at the protein level (*Figure 1—figure supplement 1D*). In the Broad Institute's Cancer Dependency Map (DepMap; *Tsherniak et al., 2017*), of all the FABPs, only *FABP5* exhibited a negative CERES Score (–0.30) in all 20 MM cell lines, demonstrating a strong reliance on *FABP5* for their survival (*Figure 1—figure supplement 2A*). Interestingly, all cancer types within the DepMap database had negative *FABP5* CERES values (*Figure 1—figure supplement 2B*). Importantly, many fatty acid metabolism genes, including *FABP5*, had negative CERES scores (shown in blue) in MM cells (*Figure 1—figure supplement 2C*).

Based on these initial findings, we next examined the effect of *FABP5* knockout (KO) in MM cells. *FABP5* KO (*FABP5*KO) MM.1R cells exhibited a 94% editing efficiency with a ~59% KO efficiency after expansion (*Figure 1—figure supplement 3A and B*). We observed an 84% reduction in *FABP5* expression in the edited pool (*Figure 1—figure supplement 3C*), confirming functional *FABP5* knockdown. *FABP4* expression was not altered (*Figure 1—figure supplement 3D*), but *FABP6* expression was increased in the edited cells (*Figure 1—figure supplement 3E*). *FABP5* KO cells showed a slight reduction in cell numbers at 48, 72, and 96 hr, versus controls (*Figure 1—figure supplement 3F*).

### Pharmacological inhibition of FABPs reduces myeloma cell proliferation in vitro

Having observed potential compensation among FABP family members in the *FABP5*KO cells, we next used two well-known FABP inhibitors (FABPi): BMS309403 and SBFI-26, which specifically and potently inhibit FABPs by binding their canonical ligand-binding pockets, or inducing conformational changes, for example by binding their substrate entry portal region (*Hsu et al., 2017*). Ligand-binding assays determined that BMS309403 has $K_i$ values in solution of <2, 250, and 350 nM for FABP4, FABP3, and FABP5, and that SBFI-26 has $K_i$ values of 900 and 400 nM for FABP5 and FABP7, respectively, as reported on the manufacturers' datasheets (*Hsu et al., 2017*). BMS309403 and SBFI-26 consistently demonstrated dose-dependent decreases in myeloma cell numbers, in all 7 MM lines screened, at 72 hr (*Figure 1C and D*; *Supplementary files 2 and 3*) and earlier (*Figure 1—figure supplement 4*). BMS309403 (50 μM), SBFI-26 (50 μM), or the combination (50 μM BMS309403 +50 μM SBFI-26) reduced cell numbers at 24, 48, and 72 hr by 39%, 42%, and 83%, respectively in GFP+/Luc+MM.1S cells (*Figure 1E*), suggesting that targeting different FABPs, or using different FABP inhibitors, could be beneficial. Non-cancerous cells were much less sensitive to FABPi (*Figure 1F*), intimating the potential clinical translation of these or similar FABP inhibitors, as supported by prior literature showing the safety of FABP inhibitors (*Al-Jameel et al., 2017*; *Mukherjee et al., 2020*). No change in amount or localization of FABP5 protein after treatment with FABPi was observed by immunofluorescence (*Figure 1—figure supplements 5 and 6*) at 24 hr in GFP+/Luc+MM.1S or RPMI-8226 cells, or by western blotting at 24, 48, or 72 hr in GFP+/Luc+MM.1S cells (*Figure 1—figure supplement 7A and B*). Gene expression of *FABP3*, *FABP4*, *FABP5*, and *FABP6* were also not consistently altered with treatments (*Figure 1—figure supplement 7C*) as assessed by qRT-PCR. These data suggest that FABP

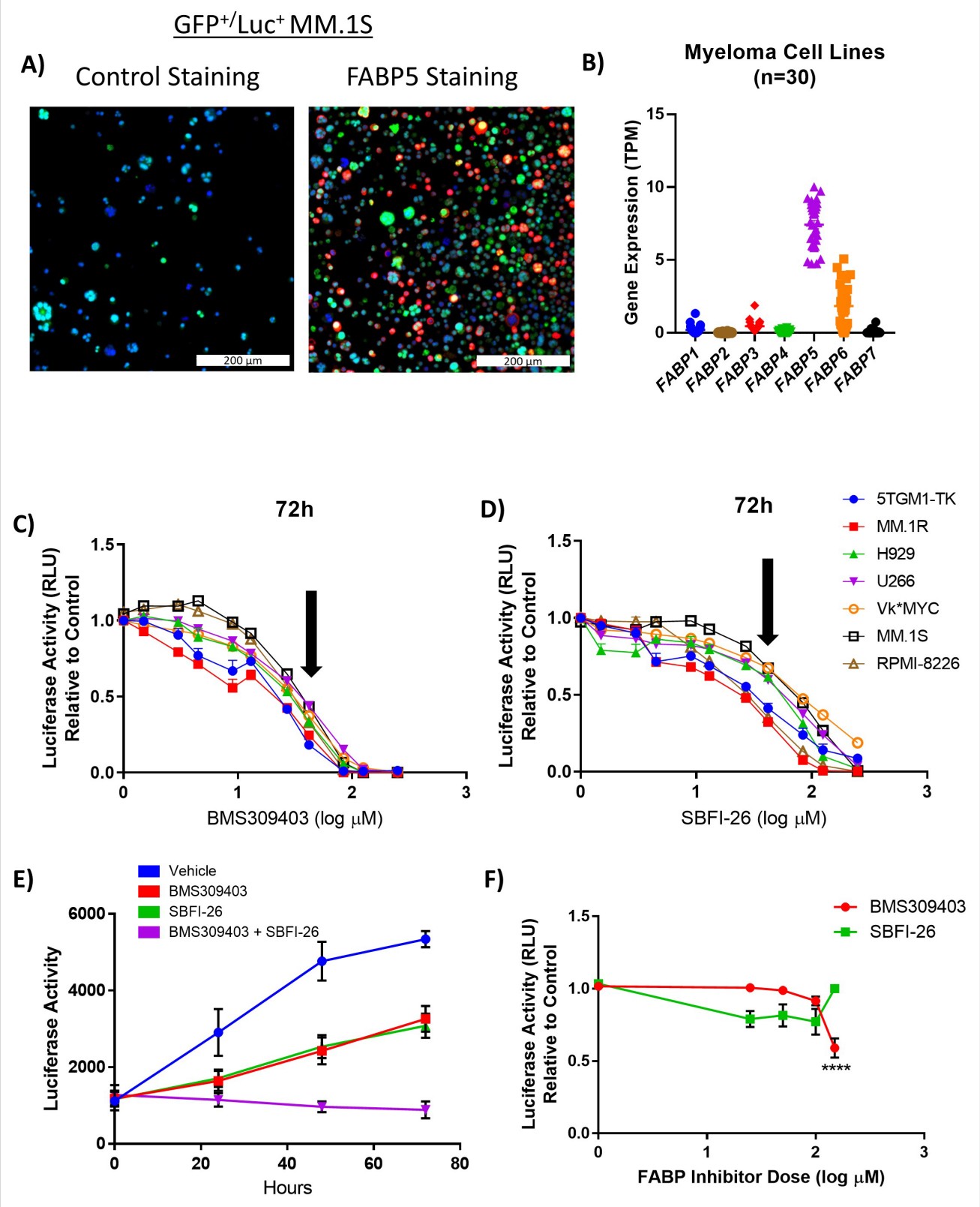

**Figure 1.** FABPi significantly impair MM cell growth and induces apoptosis. (**A**) Confocal overlay immunofluorescence images show FABP5 (red) expressed in cytoplasm of GFP+/Luc+ MM.1S cells. Nuclei identified with DAPI (blue), cells stained with secondary antibody alone (control) or primary plus secondary antibodies (FABP5 staining), scale bar = 200 μm. (**B**) Comparison of basal gene expression of FABP isoforms in 30 myeloma cell lines. Data extracted from the Cancer Cell Line Encyclopedia (CCLE; DepMap, Broad (2022): DepMap 22Q2 Public. figshare. dataset. https://doi.org/10.6084/

*Figure 1 continued on next page*

*Figure 1 continued*

[m9.figshare.19700056.v2](m9.figshare.19700056.v2)), filtered in excel, and graphs made in Graphpad PRISM (v7.04) using scatter dot plots (mean ± SEM). (**C, D**) MM cell numbers after being exposed to (**C**) BMS309403 and (**D**) SBFI-26 for 72 hr; 50 µM dose (~EC50) indicated by arrows. (**E**) GFP+/Luc+MM.1S cell numbers after treatment with inhibitors in combination (50 µM each). Vehicle vs BMS309403 (24 hr, *; 48 hr, ****; 72 hr, ****). Vehicle vs SBFI-26 (24 hr, *; 48 hr, ****; 72 hr, ****). Vehicle vs BMS309403 +SBFI-26 (24 hr, ***; 48 hr, ****; 72 hr, ****). BMS309403 vs BMS309403 +SBFI-26 (48 hr, **; 72 hr, ****). SBFI-26 vs BMS309403 +SBFI-26 (48 hr, **; 72 hr, ****). Two-way ANOVA analysis with Tukey's multiple comparisons test analysis. (**F**) CellTiter-Glo analysis of human mesenchymal stem cells after treatment with BMS309403 or SBFI-26 for 72 hr. Data are mean ± SEM and represent averages or representative runs of at least three experimental repeats. One-way ANOVA with Dunnett's multiple comparison test significance shown as *p<0.05. **p<0.01. ***p<0.001. ****p<0.0001. **** p<0.0001. Please see 8 supplements to Figure 1.

The online version of this article includes the following figure supplement(s) for figure 1:

**Figure supplement 1.** Expression of FABP family members in MM and hematopoetic/lymphoid cell lines.

**Figure supplement 2.** FABP Depmap CERES scores in tumor cells show that FABP proteins are clinically relevant in MM.

**Figure supplement 3.** *FABP5* knockout (KO) MM.1R myeloma cell line.

**Figure supplement 4.** FABP inhibitors exhibit consistent negative effects on cell number in eight myeloma cell lines.

**Figure supplement 5.** FABP inhibitor treatment did not induce changes in amount or localization of FABP5 in GFP+/Luc+MM.1 S cells.

**Figure supplement 6.** FABP inhibitor treatment did not induce changes in amount or localization of FABP5 in RPMI-8226 myeloma cells.

**Figure supplement 7.** Protein levels of FABP5 are not affected by inhibitors in MM.1S cells.

**Figure supplement 8.** Treatment with recombinant FABP4 or FABP5 has no effect on GFP+/Luc+ MM.1 S cell numbers.

activity, but not protein expression, is decreased by these FABP inhibitors. Recombinant FABP4 and FABP5 did not affect MM.1S cell number (*Figure 1—figure supplement 8A,B*).

## FABPi induce gene expression changes in myeloma cells that affect a range of cellular processes and pathways linked to survival

To identify transcriptional changes that may mediate the effects of FABP inhibition on cell number, we treated GFP+/Luc+MM.1S cells with a vehicle control, the single FABP inhibitors alone (50 µM), or the combination of FABPi (50 µM of each) for 24 hr in vitro, isolated total RNA, and performed RNA-Seq. Principal component analysis (PCA) demonstrated that the FABP inhibitor groups exhibited distinct gene expression profiles, and that the combination treatment differed the most from vehicle-treated cells (*Figure 2A*). Over 14,000 genes were analyzed, revealing 93 significant differentially expressed (DE) genes within all three treatment groups, compared to the vehicle control (FDR <0.2): 90 down-regulated and 3 upregulated (*Figure 2B*; *Supplementary file 4*). Consistent with decreased levels of transcription, we also observed significantly lower levels of 5-hydroxymethylcytosine in cells treated with FABPi compared to vehicle-treated cells (*Figure 2C*), suggesting decreases in active chromatin. This finding is consistent with previous reports linking FABP depletion to DNA methylation signatures in other cancers (*Mukherjee et al., 2020*; *Yan et al., 2018*).

To further understand the mechanisms of action of FABPi, we investigated which pathways were impacted in our RNA-Seq data using STRINGdb and IPA (Ingenuity Pathway Analysis). IPA was specifically used to investigate canonical pathways, while STRINGdb was used to examine connectivity of DE genes and enrichment for specific gene ontology terms, as well as molecules in Reactome and KEGG pathways. In total, 15 IPA canonical pathways were commonly dysregulated in all three treatment groups including Cell Cycle: G2/M DNA Damage Checkpoint Regulation, EIF2 Signaling, Sirtuin Signaling Pathway, and the NER pathway (*Figure 2D*; *Supplementary file 5*). The one upregulated pathway according to STRING was 'cellular response to interferon gamma signaling' in the combination group (*Figure 2E*; *Supplementary file 6*). The top downregulated pathways in the combination treatment by STRING analysis are in *Supplementary file 7*.

Interestingly, both IPA and STRING databases revealed commonly downregulated pathways related to the unfolded protein response (UPR) or ER stress responses for BMS309403 (*Figure 2—figure supplement 1A–C*), SBFI-26 (*Figure 2—figure supplement 3*, and the combination *Figure 2D and F*). Three of the five downregulated Reactome pathways in the combination group were related to UPR or ER stress (*Figure 2F*), driven by molecular players such as *XBP1*, BIP (*HSPA5*), and IRE1 (*ERN1*) (*Figure 2—figure supplement 3A*). Downregulation of total *XBP1* by the combination treatment was confirmed after 24 hr (*Figure 2—figure supplement 3B*) and heatmaps visually demonstrated the downregulation of genes involved in XBP1 signaling (*Figure 2—figure supplement 3C*) and the UPR

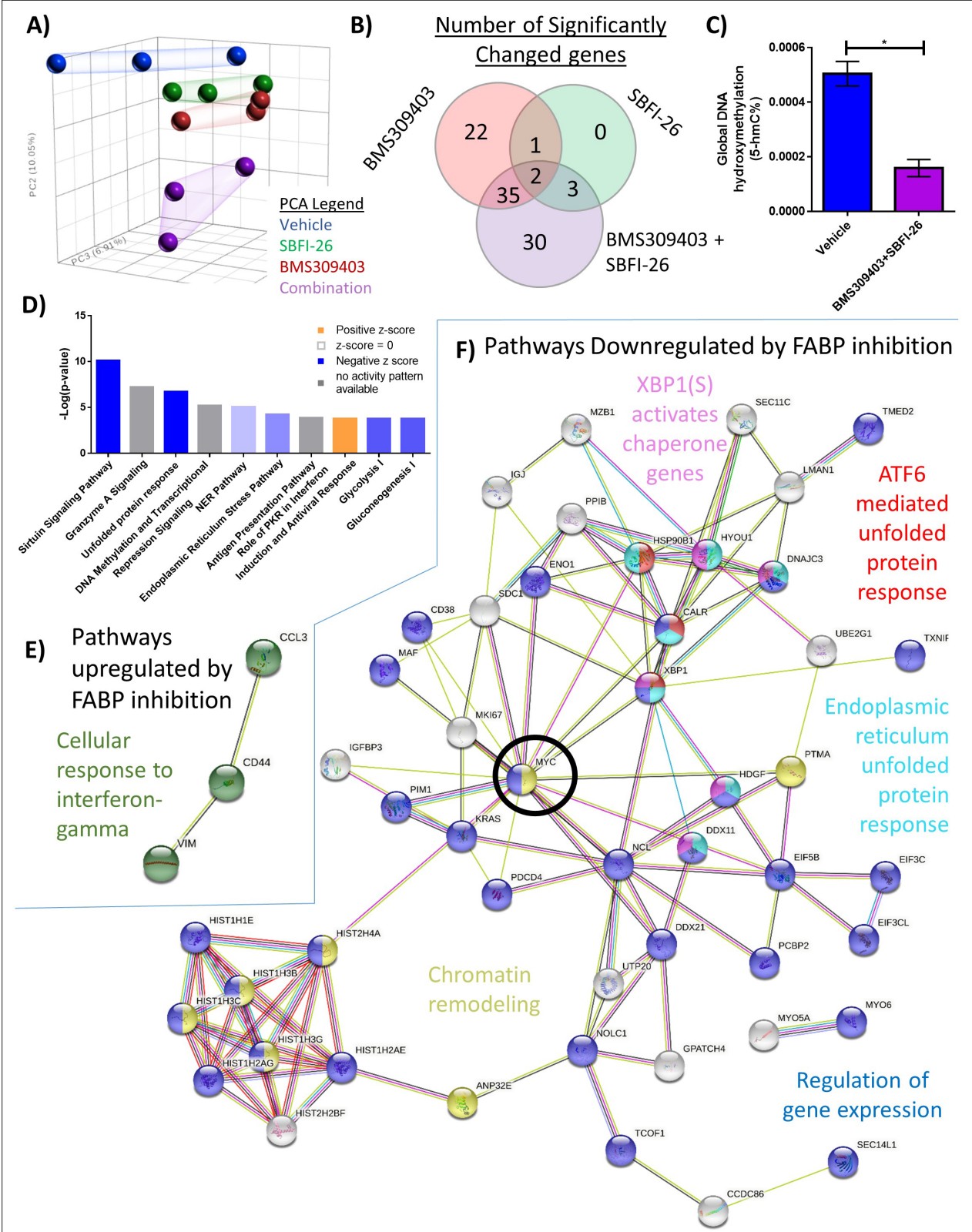

**Figure 2.** RNA sequencing analysis of MM1S cells treated with FABPi for reveals unique gene expression patterns. (**A**) Principal component analysis of cells after 24 hr treatments. (**B**) Venn diagram displays the overlapping and specific genes dysregulated with FABPi (FDR cutoff of 0.2). (**C**) Global hydroxymethylation DNA analysis of MM.1S cells after 24 hr of combination treatment. Data represent mean and +/- SEM using n=3 biological repeats, and * $p < 0.05$ using an unpaired, two-tailed Student t-test. (**D**) Ingenuity pathway analysis of RNA-Seq results (p-value of overlap by Fisher's exact

*Figure 2 continued on next page*

Figure 2 continued

test, significance threshold value of p<0.05(-log value of 1.3)). Stringdb (FDR cutoff of 0.2) of the combination therapy versus control showing (**E**) the 1 upregulated pathway and (**F**) 5 of the many downregulated pathways. MYC, a central node, is circled for emphasis. GFP+/Luc +MM.1 S cells were used for these experiments. Please see 3 supplements to Figure 2.

The online version of this article includes the following figure supplement(s) for figure 2:

**Figure supplement 1.** RNA sequencing analysis of GFP+/Luc+MM.1 S cells treated for 24 hr with BMS309403.

**Figure supplement 2.** RNA sequencing analysis of GFP+/Luc+MM.1 S cells treated for 24 hr with SBFI-26.

**Figure supplement 3.** RNA sequencing analysis of GFP+/Luc+MM.1 S cells treated with FABP inhibitors reveals a unique gene expression suggesting endoplasmic reticulum stress.

(*Figure 2—figure supplement 3D*) as determined by IPA. Interestingly, *MYC*, a known oncogene, was a central node in STRING analysis (*Figure 2F*) and among the top 10 most downregulated genes in RNA-Seq from combination treatments (*Supplementary file 8*).

## FABPi induces protein changes in MM cells that affect a range of cellular processes and pathways linked to survival

To identify protein changes resulting from FABPi, we treated GFP+/Luc+MM.1S cells with the single inhibitors (50 µM) or the combination (50 µM of each) for 48 hr, isolated total cell lysate proteins, and performed a mass spectrometry-based proteomic analysis. (Numbers of significant proteins, *Supplementary file 9*; protein names, *Supplementary files 10-15*). PCA analysis showed a tight grouping of samples based on treatments (*Figure 3—figure supplement 1A*); 15 proteins were commonly upregulated and 15 were commonly downregulated between all treatments (*Figure 3—figure supplement 1B*, C; *Supplementary files 16 and 17*).

We then compared significant genes and proteins identified by both RNA-Seq and proteomics (*Figure 3A and B*). CCL3, a chemokine for monocytes, macrophages, and neutrophils, was upregulated by SBFI-26, BMS309403, and their combination in proteomics, and upregulated by the combination treatments in RNA-Seq. Ki67, a proliferation marker, and PTMA, a negative regulator of apoptosis, were both significantly downregulated in the combination treatment in RNA-Seq and proteomics, and in the single drug treatments in proteomics (*Figure 3B*), indicating cell death and cell cycle arrest likely result from FABPi.

STRING analysis of proteomic data suggested many other systemic changes (eg, downregulation of DNA replication and other viability/proliferation processes and upregulation of lysosome, carboxylic acid catabolic process, and mitochondrial pathways) induced by the FABPi combination treatments (*Figure 3C and D*). STRING analysis also revealed interesting up- and downregulated pathways by BMS309403 or SBFI-26 treatments alone (*Figure 3—figure supplement 2*, *Figure 3—figure supplement 3*). IPA analysis revealed 'EIF2 Signaling' to have the highest negative Z-score for all FABPi treatments in proteomics (*Figure 3E*; *Figure 3—figure supplement 4A*, *Figure 3—figure supplement 5A*). IPA 'Cell Death and Survival' heatmap analysis showed increases in cell death and apoptosis pathways and decreases in cell viability pathways after FABPi combination treatment (*Figure 3F*; *Figure 3—figure supplements 4B and 5B*). Interestingly, MYC was the most significant predicted upstream regulator, found to be strongly inhibited in the BMS309403, SBFI-26, and combination treatments from IPA proteomic analysis (*Supplementary files 18-20*).

Since *MYC* was found as a central node or commonly downregulated gene/pathway in our RNA-Seq and proteomic data analyses, we investigated MYC's role in FABP signaling in myeloma cells. We confirmed decreased *MYC* expression in GFP+/Luc+MM.1S cells treated with the FABPi combination, and also saw a trend for this in 5TGM1-TK cells treated with SBFI-26 (*Figure 3—figure supplement 6A*, B). MYC protein level was also decreased in GFP+/Luc+MM.1S cells at 24, 48, and 72 hr with FABPi (*Figure 4A and B*), with similar trends observed in 5TGM1-TK myeloma cells (*Figure 3—figure supplement 6C*, D). The decrease in MYC-regulated genes with FABPi was also visualized in both the RNA-Seq (*Figure 4C*) and proteomic data (*Figure 4D*) by heatmap analysis. In RNA-Seq data, treatment with BMS309403 induced aberrant gene expression of 171 genes known to be regulated by MYC (*Supplementary file 21*), with 138 of those having expression patterns consistent with MYC inhibition. Similarly, co-treatment induced changes in 91 genes modulated by MYC

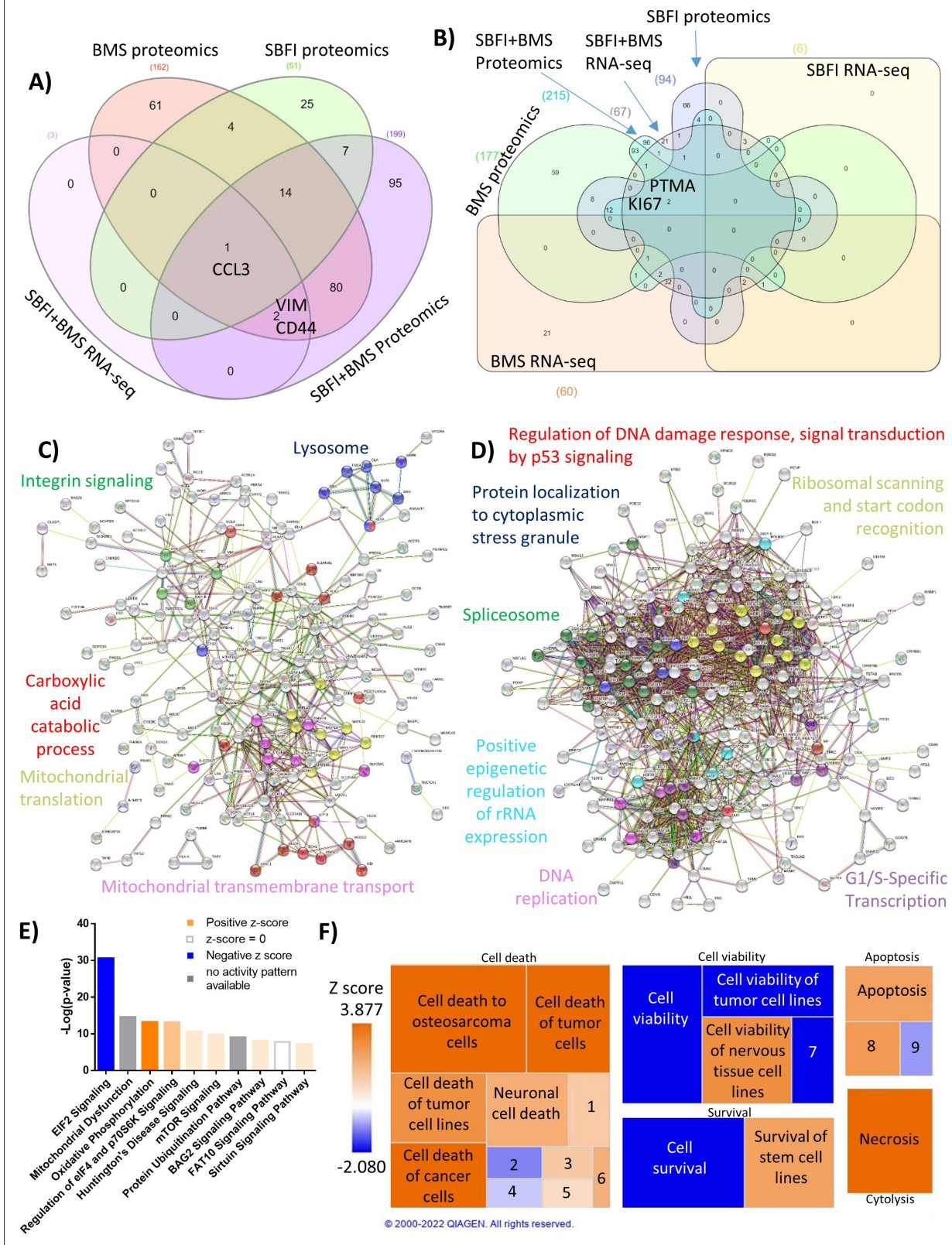

**Figure 3.** Forty-eight hr proteomic analysis of MM1S cells treated with FABPi reveals a unique protein signature. MM.1S cells were assessed by proteomics after 48 hr treatments with BMS309403 (50 μM), SBFI-26 (50 μM) or the combination, and compared to results from RNA-Seq. N=3 biological replicates and three technical replicates Venn diagram comparison of (**A**) upregulated genes and (**B**) downregulated proteins in proteomics and RNA-Seq among BMS309403 and SBFI-26 treated cells compared to vehicle. (**C–F**) Pathway analysis of proteomic data of significantly upregulated or

*Figure 3 continued on next page*

Figure 3 continued

downregulated proteins in MM.1S cells treated with both FABPi (BMS309403 +SBFI-26). (**C, D**) String analysis of upregulated (**C**) or downregulated (**D**) pathways. (**E**) Top 10 significantly changed pathways with FABP inhibition. For IPA analysis, orange represents positive z-score, blue indicates a negative z-score, gray represents no activity pattern detected and white represents a z-score of 0. (**F**) Ingenuity pathway analysis of the Cell Death and Survival heatmap. Numbers in boxes represent: (1) Cell death of melanoma lines; (2) Cell death of carcinoma cell lines; (3) Cell death of neuroblastoma cell lines; (4) Cell death of breast cancer cell lines; (5) Cell death of connective tissue cells; (6) Cell death of fibroblast cell lines; (7) Cell viability of myeloma cell lines; (8) Apoptosis of tumor cell lines; (9) Apoptosis of carcinoma cell lines. GFP+/Luc +MM.1 S cells were used for these experiments. Please see 7 supplements to Figure 3.

The online version of this article includes the following figure supplement(s) for figure 3:

**Figure supplement 1.** Mass spectrometry analysis revealed 48 hr treatment with FABP inhibitors induces a significant change in proteomic profile.

**Figure supplement 2.** Mass spectrometry analysis reveals a shift in the proteomic profile of GFP⁺/Luc⁺ MM.1 S cells treated with BMS309403 for 48 hours: STRING analysis.

**Figure supplement 3.** Mass spectrometry analysis reveals a shift in the proteomic profile GFP⁺/Luc⁺ MM.1 S cells treated with SBFI-26 for 48 hr: STRING Analysis.

**Figure supplement 4.** Mass spectrometry analysis reveals a shift in the proteomic profile of GFP⁺/Luc⁺ MM.1 S cells treated with BMS309403 for 48 hr: IPA analysis.

**Figure supplement 5.** Mass spectrometry analysis reveals a shift in the proteomic profile of GFP⁺/Luc⁺ MM.1 S cells treated with SBFI-26 for 48 hr: IPA Analysis.

**Figure supplement 6.** FABP inhibitor treatment alters expression of *MYC* gene and MYC protein expression and MYC-regulated genes.

**Figure supplement 7.** Co-treatment with BMS309403 and SBFI-26 induced changes in 91 genes modulated by MYC in GFP⁺/Luc⁺ MM.1 S cells based on RNA-seq data.

(*Figure 3—figure supplement 7*; 68 consistent with MYC downregulation), while 29 MYC targets were aberrantly expressed with SBFI-26 treatment (*Figure 2—figure supplement 2D*; 18 consistent with MYC downregulation).

To test if MYC inhibition was a major cause of the FABPi effects on MM cells, we then pharmacologically inhibited MYC and tested a range of doses of FABPi. MYC inhibition alone dramatically reduced cell numbers at 72 hr, as expected, and FABP inhibition had less of an effect on MM cells when MYC was already inhibited (seen by a slope of ~0 for the black lines) (*Figure 4E and F*). This suggests that much of the effect of FABPi is through decreased MYC signaling, although the strong effect of the MYC inhibitor makes this difficult to determine unhesitantly. Similar results were seen at 24 and 48 hr (*Figure 4—figure supplement 1*).

## FABPi impair MM cell metabolism, mitochondrial function, and cell viability

Having observed effects of the inhibitors on metabolic processes such as mitochondrial function and oxidative phosphorylation in the proteomic data, we next assessed mitochondrial function and metabolic changes using a Cell Mito Stress Test (*Figure 5—figure supplement 1A*). After 24 hr treatments, all FABPi treatments decreased basal mitochondrial oxygen consumption rates (OCR) and OCR dedicated to ATP production (*Figure 5—figure supplement 1B*). Maximal respiration and spare respiratory capacity were decreased with SBFI-26 and combination treatments, suggesting FABP inhibition reduces the ability of MM cells to meet their energetic demands.

To determine the effects of FABPi on fatty acid oxidation (FAO) specifically, we treated tumor cells with etoxomir, an FAO inhibitor, with or without the combination FABPi treatment (*Figure 5—figure supplement 2*). The combination of FABPi alone again strongly reduced mitochondrial respiration in most of the parameters assessed. Interestingly, etoxomir treatment caused a slight, but significant reduction in OCR when it was administered, demonstrating some reliance of MM cells on FAO for mitochondrial respiration. However, the FABPi had a much greater effect on MM mitochondrial respiration than etoxomir alone, suggesting that FABPi treatment inhibited mitochondrial respiration through another mechanism. Also, since maximal respiration was decreased in the Etox +FABPi combination compared to FABPi alone, it appears that FABPi treatment does not completely block FAO when used alone. Overall, the data demonstrate that mitochondrial respiration is inhibited by FABPi. To assess whether metabolic dysfunction could be caused by damaged mitochondria, we utilized tetramethylrhodamine, ethyl ester (TMRE) staining and flow cytometric analysis to assess

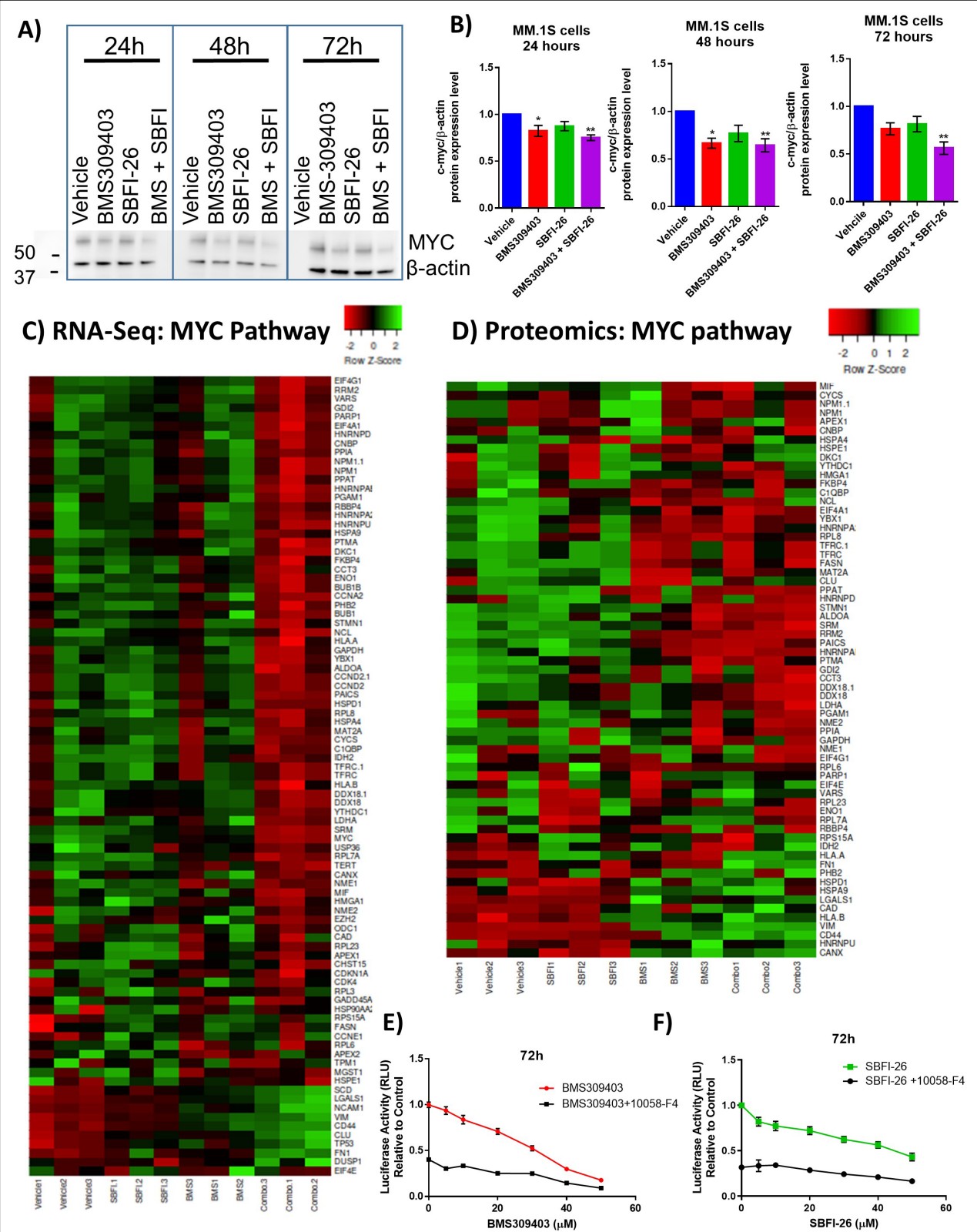

**Figure 4.** FABPi target MYC and the MYC pathway. (**A**) Representative western blot and (**B**) quantification of MYC protein and β-actin (housekeeping control) at 24, 48, and 72 hr after treatment with BMS309403 (50 μM), SBFI-26 (50 μM), or the combination. (**C**) RNA-seq and (**D**) Proteomic analysis of expression of genes/proteins involved in MYC signaling shown as heatmap visualizations. Curated lists are based on IPA MYC Pathway list, known MYC-regulated genes, and proteins present in proteomics. (**E**) 72 hr BMS309403 dose curve with and without Myc inhibitor 10058-F4 (37.5 μM) in MM.1S cells.

*Figure 4 continued on next page*

*Figure 4 continued*

(**F**) 72 hr SBFI-26 dose curve with and without 10058-F4 (37.5 µM) in MM.1S cells. Data represent mean ± SEM from n=3 biological repeats, analyzed with one-way ANOVA with significance shown as *p<0.05. **p<0.01. ****p<0.0001. GFP+/Luc +MM.1 S cells were used for these experiments. Please see 1 supplement to Figure 4.

The online version of this article includes the following figure supplement(s) for figure 4:

**Figure supplement 1.** FABP inhibitors do not synergize with MYC inhibitor, 10058-F4, in GFP⁺/Luc⁺ MM.1 S cells at 24 and 48 hr.

mitochondrial transmembrane potential. GFP⁺/Luc⁺MM.1S cells treated with BMS309403 or the combination (BMS309403 +SBFI-26) had decreased TMRE staining (*Figure 5—figure supplement 3*), suggesting that BMS309403 damages MM cell mitochondria.

We next investigated if reactive oxygen species (ROS), a major byproduct of the electron transport chain, were changing in MM cells after FABPi treatment. CellROX staining showed that the combination FABPi treatment significantly increased total ROS at 24, 48, or 72 hr in MM.1S (ATCC), U266 and OPM2 cells (*Figure 5A*, *Figure 5—figure supplements 4A and 5A*, 6 A). We also found changes in superoxide, a ROS subspecies measured by MitoSOX, after FABP inhibition; in MM.1S (ATCC), BMS309403 and the FABPi combination increased superoxides over 72 hr (*Figure 5B*, *Figure 5—figure supplement 4B*). In U266, the FABPi combination increased superoxides at each time point, and BMS309403 increased superoxides at 48 and 72 hr (*Figure 5—figure supplement 5B*). In OPM2, all FABPi treaments increased superoxides at all timepoints (*Figure 5—figure supplement 6B*). Overall, FABP proteins are vital to MM cells for normal oxygen consumption, mitochondrial potential maintenance and ATP production, adaption to increased demands for energy, and control of ROS, including superoxides.

We next investigated FABP inhibitor effects on MM cell cycle and apoptosis. In GFP+/Luc +MM.1 S, FABPi combination treatment increased the G0/G1 population at 24, 48, and 72 hr, and decreased G2/M at 48 and 72 hr, suggesting a G0/G1 arrest and a negative impact on cell cycle progression (*Figure 5C*, *Figure 5—figure supplement 7*). FABPi combination treatment also increased apoptosis in GFP+/Luc +MM.1 S cells at all three time points, and SBFI-26 did as well at 72 hr (*Figure 5D*). To determine if effects of the combination treatment were reflective purely of a higher level of inhibition, or a synergism of the different FABP inhibitors, we assessed apoptosis, cell cycle, and proliferation using a range of doses and FABP inhibitor combinations (*Figure 5—figure supplements 8 and 9*). Interestingly, in GFP+/Luc +MM.1 S, 100 µM of BMS309403 induced larger impacts on apoptosis, cell cycle arrest, and Ki67 expression than all other treatments (*Figure 5—figure supplement 8*) suggesting it may be more effective than SBFI-26 in this cell line. In RPMI-8226 cells, apoptosis and cell cycle arrest were also induced with the combination or single inhibitors (*Figure 5—figure supplement 9*). Interestingly, in this cell line, 100 µM of single inhibitors elicited similar responses to combination treatment inhibitors (50 µM BMS309403 +50 µM SBFI-26), suggesting that FABP inhibitors may have slightly different efficacies in different MM cells. We subsequently investigated the combination of FABPi with dexamethasone, a first-line therapy for MM patients. Dexamethasone and FABPi showed promising, additive effects on cell numbers and apoptosis in GFP⁺/Luc⁺MM.1S, OPM2, and RPMI-8226 cells (*Figure 5—figure supplement 10*), suggesting a potential to combine FABP inhibition with current therapies. In summary, FABPi treatment in vitro elicited multitudinous changes in MM cell transcriptomes and proteomes, resulting in alterations in cell cycle progression, cell viability, apoptosis, MYC signaling, cellular metabolism.

## FABPi has variable effects on tumor burden and survival in myeloma mouse models

To investigate the efficacy of FABPi in vivo, we utilized two murine myeloma models. First, we examined the efficacy of FABPi in the GFP⁺/Luc⁺MM.1S SCID-beige xenograft model. Treatments began with 5 mg/kg BMS309403, 1 mg/kg SBFI-26, the combination, or vehicle 3 X/week (*Figure 6—figure supplement 1A*) one day after GFP⁺/Luc⁺MM.1S tail vein inoculation. Bone mineral density (BMD), but not bone mineral content (BMC), was slightly lower after BMS309403 treatment (*Figure 6—figure supplement 1B*, C), although this group also started with a slightly lower BMD, and fat mass, but not lean mass was decreased with the combination treatment (*Figure 6—figure supplement 1D*, E). FABPi did not influence mouse weight (*Figure 6A*), but a difference in tumor burden assessed by BLI

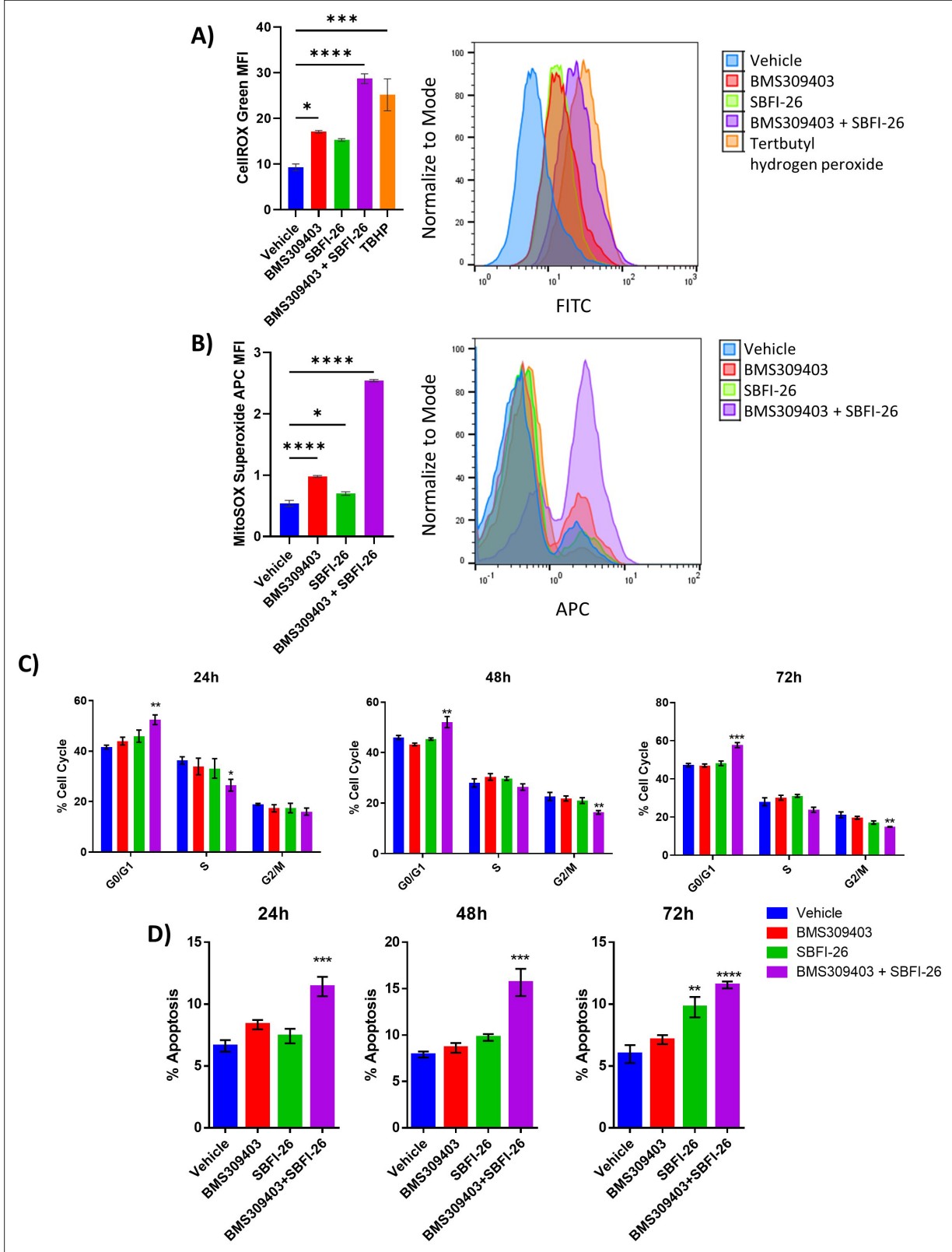

**Figure 5.** FABPi significantly induce reactive oxygen species, impair MM cell growth and induce apoptosis. (**A**) Reactive oxygen species measured by MFI (mean fluorescent intensity) with CellROX Green staining at 72 hr in MM.1S cells. TBHP is positive control. (**B**) Superoxide levels shown as MFI, determined with MitoSOX staining, at 72 hr in MM.1S cells. (**C**) MM.1S cell cycle states with the FABPi alone (50 μM) or in combination (50 μM of each). (**D**) Apoptosis in MM.1S cells with FABPi as in C. Data are mean ± SEM unless otherwise stated and represent averages or representative runs of at

*Figure 5 continued on next page*

*Figure 5 continued*

least three experimental repeats. One-way ANOVA with Dunnett's multiple comparison test significance shown as *p<0.05. **p<0.01. ***p<0.001. ****p<0.0001. ATCC MM.1S cells were used for these experiments. Please see 10 supplements to Figure 5.

The online version of this article includes the following figure supplement(s) for figure 5:

**Figure supplement 1.** FABP blockade reduces cell metabolism: (**A, B**) GFP$^+$/Luc$^+$MM.1S cells were cultured for 24 hr with BMS309403 (50 µM), SBFI-26 (50 µM), or both and then plated for Seahorse XF96 analysis in 96-well format.

**Figure supplement 2.** FABP blockade reduces fatty acid oxidation: (**A,B**) GFP$^+$/Luc$^+$MM.1S cells were cultured for 24 hr with BMS309403 (50 µM) plus SBFI-26 (50 µM) and then plated for Seahorse XF96 analysis in 96-well format.

**Figure supplement 3.** TMRE staining reveals compromised mitochondria in response to BMS309403 and the combination treatment.

**Figure supplement 4.** CellROX and Mitosox staining reveals an increase in total ROS and superoxide over 48 hr treatment with BMS309403 and combination therapy in MM.1S cells.

**Figure supplement 5.** CellROX and Mitosox staining reveals an increase in total ROS and superoxide over 72 hr treatment with BMS309403 and combination therapy in U266 cells.

**Figure supplement 6.** CellROX and Mitosox staining reveals an increase in total ROS and superoxide over 72 hr treatment with BMS309403 and combination therapy in OPM2 cells.

**Figure supplement 7.** MM.1S cell cycle gating after FABP inhibitor treatment.

**Figure supplement 8.** Apoptosis, cell cycle arrest and reduction in Ki67 expression is induced in GFP$^+$/Luc$^+$ MM.1 S cells through inhibition of FABP proteins at 72 hr.

**Figure supplement 9.** Apoptosis, cell cycle arrest and reduction in Ki67 expression is induced in RPMI-8226 cells through inhibition of FABP proteins at 72 hr.

**Figure supplement 10.** The effects of FABP inhibitors combined with dexamethasone after 72 hr in vitro.

was detected at day 21 with all FABPi versus vehicle, and this difference continued throughout the study (**Figure 6B and C**). Consistent with reduced tumor burden, mice receiving FABPi survived longer than the vehicle-treated mice (**Figure 6D**). Similarly, in the GFP$^+$/Luc$^+$ 5TGM1-TK/KaLwRij syngeneic model (**Figure 6—figure supplement 2A**), mice treated with 5 mg/kg BMS309403 showed increased survival (**Figure 6E**) without significant body weight changes (**Figure 6—figure supplement 2B**). However, due to variable responses to different doses of FABP inhibitors in mice of different ages (publication in preparation), we repeated the GFP$^+$/Luc$^+$MM.1S SCID-Beige study. As in our first study, mice gained weight over the course of the study with no treatment effect (**Figure 7A**). However, in this cohort, treatments had on slight, non-significant effects on tumor burden (**Figure 7B and C**), and no effect on survivial (**Figure 7D**). The in vivo data thus demonstrate a need to explore and identify factors currently limiting the efficacy of these FABP inhibitors in vivo.

## Elevated expression of *FABP5* in MM cells corresponds to worse clinical outcomes for patients

To establish potential clinical relevancy, we next tested for an association between FABP5 and MM in independent patient datasets using Multiple Myeloma Research Foundation (MMRF) CoMMpass and OncoMine. In the CoMMpass database, ~70% of myeloma patient cases exhibited moderate-to-high expression of *FABP5* (defined as >10 counts; **Figure 8—figure supplement 1A**). *FABP3*, *FABP4*, and *FABP6* were expressed by MM cells at lower levels (**Figure 8—figure supplement 1A**, insert). We next tested for an association between FABP5 and MM in independent microarray datasets using Onco-Mine. The Zhan dataset indicated that patients with higher MM cell *FABP5* expression had significantly shorter overall survival (OS) than those with lower expression (**Zhan et al., 2006**), (**Figure 8A and B**), which was confirmed in the Mulligan dataset (**Mulligan et al., 2007**, **Figure 8C**). Similarly, the Carrasco dataset showed a shorter progression-free survival (PFS) in MM patients with high versus low *FABP5* expression (**Figure 8D**, **Carrasco et al., 2006**). Moreover, patients of the high-risk/poor prognosis subtype had higher *FABP5* expression than those in the more favorable subtypes (**Zhan et al., 2006**, **Figure 8E**). In the Chng dataset (**Chng et al., 2007**), relapsed patients showed increased *FABP5* expression versus newly-diagnosed patients (**Figure 8F**). Worse PFS and OS in patients with elevated *FABP5* expression levels was then confirmed in the CoMMpass dataset (log-rank-value for high vs. low expression,<0.0001 for both PFS and OS; **Figure 8—figure supplement 1B**, C). In the Cox proportional hazards model, high *FABP5* expression was associated with a 64% increased risk of

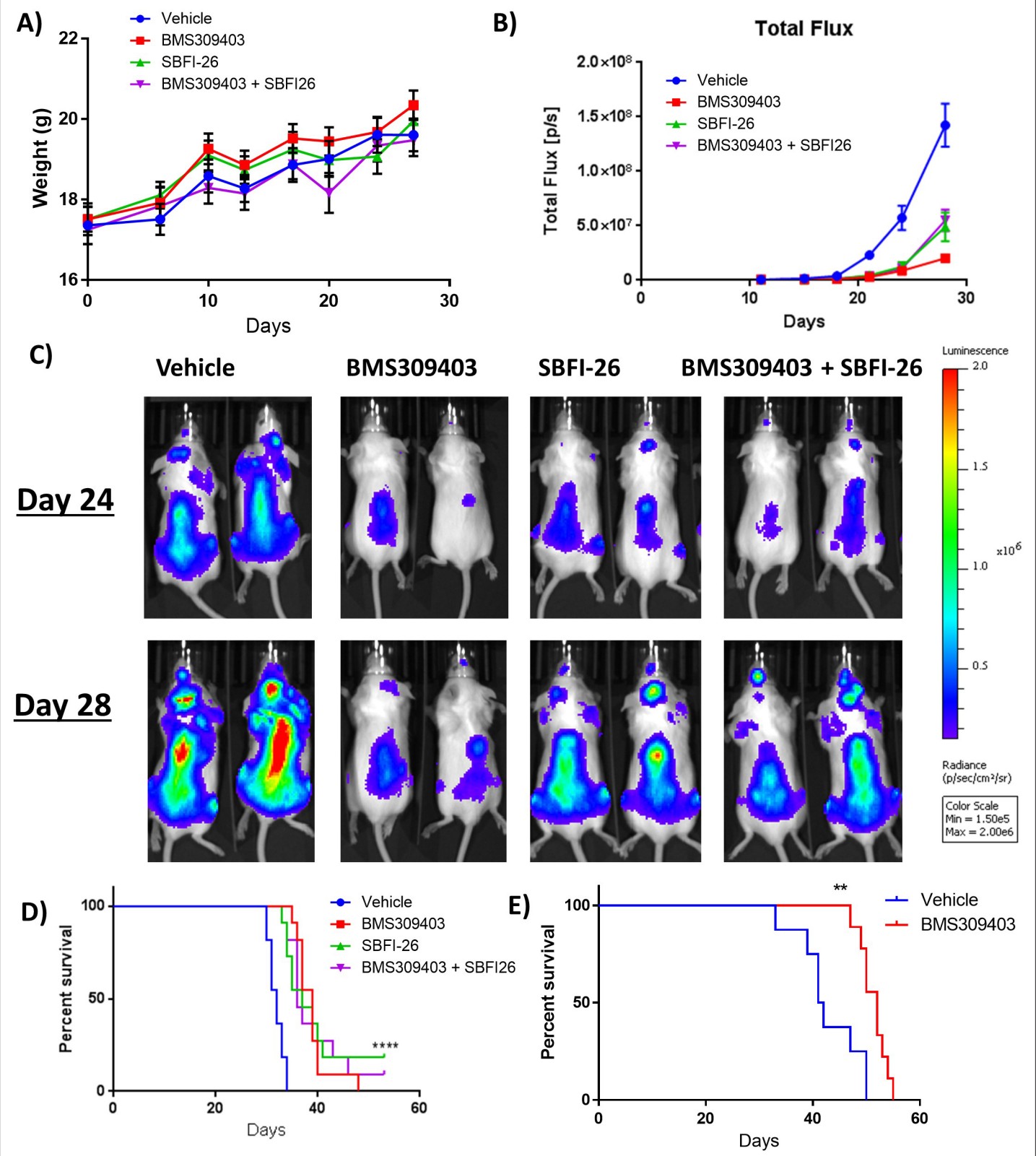

**Figure 6.** FABPi do not consistently increase survival or decrease tumor burden in myeloma xenograft (cohort 1) and syngeneic mouse models. (**A**) Mouse weights from the first cohort of SCID-beige- GFP+/Luc +MM.1 S mice treated with BMS309403, SBFI-26, or the combination from day of injection plotted as Mean ± SEM. (**B**) Tumor burden from cohort 1 of SCID-beige GFP+/Luc +MM.1 S mice assessed by bioluminescence imaging (BLI) in MM.1S model. In panel B, One-way ANOVA with Dunnett's multiple comparison test significance shown as *p<0.05. **p<0.01. ***p<0.001.

*Figure 6 continued on next page*

*Figure 6 continued*

****p<0.0001. Vehicle vs BMS309403 (24 days, ****; 28 days, ****). Vehicle vs SBFI-26 (24 days ****; 28 days, ****). Vehicle vs BMS309403 +SBFI-26 (24 hr, ****; 28 days, ****). BMS309403 vs BMS309403 +SBFI-26 (24 days NS; 28 days, ***). SBFI-26 vs BMS309403 +SBFI-26 (24 and 28 days, NS). BMS309403 vs SBFI-26 (24 hr, NS 28 days, **). (**C**) Representative BLI images from cohort 1 of SCID-Beige MM.1S$^{gfp+luc+}$ mice at days 24 and 28. (**D**) Survival of SCID-Beige MM.1S$^{luc+}$ mice from first cohort; analysis performed by Kaplan-Meier Survival Analysis, Log-Rank (Mantel-Cox) test, p<0.0001, n=11. (**E**) Survival of KaLwRij mice injected with 5TGM1 cells. Survival analysis performed by Kaplan-Meier Survival Analysis, Log-Rank (Mantel-Cox) test, p=0.0023, Vehicle n=8, BMS309403 n=9. Please see 2 supplements to Figure 6.

The online version of this article includes the following figure supplement(s) for figure 6:

**Figure supplement 1.** SCID-Beige-GFP+/Luc +MM.1 S in vivo characterization.

**Figure supplement 2.** GFP$^{+}$/Luc$^{+}$ 5TGM1-TK/KaLwRij syngeneic model.

disease progression or death (HR: 1.64; CI: 1.34, 2.00), and a twofold increased risk of early death (HR: 2.19; CI: 1.66, 2.88).

Since obesity is a known MM risk factor (*Marinac et al., 2018*) and FABP5 can regulate diet-induced obesity (*Shibue et al., 2015*), we explored the influence of body mass index (BMI) on our findings in the CoMMpass dataset. BMI was not associated with *FABP5* in a general linear model adjusting for age or sex, and the addition of BMI to the Cox model of *FABP5* expression described above did not materially attenuate the effect estimates, suggesting *FABP5* expression is a BMI-independent biomarker for MM aggressiveness. We also examined genes correlated with *FABP5* and found none ontologically related to obesity, again suggesting that FABP5 effects are BMI-independent (*Figure 8—figure supplement 1D*; *Supplementary file 22*). When all other FABPs expressed in MM cells (*FABP6*, *FABP3*, and *FABP4*) were examined, only *FABP6* also affected hazard ratios (although effect sizes were not as large as *FABP5*) for PFS (HR:1.48; CI 1.172, 1.869) and OS (HR:1.837, CI: 1.347, 2.504), indicating that it may also be a biomarker for worse outcomes (*Figure 8—figure supplement 2*). Overall, these data across multiple datasets provide rationale to explore the molecular and functional roles of the FABPs in the MM setting.

## Discussion

Herein, we describe our finding that the FABPs are a family of targetable proteins that support myeloma cells. Targeting the FABP family may be a new, efficacious method to inhibit MM progression that necessitates further investigation. FABP inhibition induced apoptosis, cell cycle arrest, and inhibition of proliferation of numerous MM cell lines in vitro, while having negligible effects on non-MM cells. In vivo, FABP inhibition caused no weight loss or other overt toxicities, supporting similar findings in other pre-clinical oncology studies (*Al-Jameel et al., 2017*; *Bosquet et al., 2018*; *Herroon et al., 2013*; *Mukherjee et al., 2020*). Further analysis and experiments are still needed (e.g. histological analysis of major organs and quantification of serum toxicity markers) before targeting FABPs can be translated to humans. Myeloma cell proliferation also decreased with genetic knockout of *FABP5*, although FABP signaling compensation may have occurred via upregulation of *FABP6*. Clinical datasets and DepMap analyses also demonstrated the importance of the FABPs, specifically *FABP5*, and perhaps *FABP6*, in MM. A recent publication also analyzed patient datasets and similarly found correlations between high FABP5 expression and worse MM patient survival, and between *FABP5* mRNA levels and different immune microenvironment properties, suggesting a role for FABP5 in immunomodulation, an important hypothesis that we have not yet further explored (*Jia et al., 2021*).

FABP inhibition decreased expression of genes and pathways related to ER stress, XBP1, and the UPR. For example, *EIF5B* was downregulated by all FABPi in proteomic analysis and RNA-Seq. *EIF5B* is a translation initiation factor that promotes the binding of subunits and antagonizes cell cycle arrest via modulations of p21 and p27, and depletion of *EIF5B* could contribute to activation of ER stress (*Ross et al., 2019*). eIF5B has been implicated as a oncoprotein that aids in managing ER stress and evading apoptosis (*Ross et al., 2019*). Myeloma cells constitutively activate the UPR to protect themselves from ER stress-induced death that would otherwise result from the continuous production and secretion of immunoglobulins. Therefore, the inhibition of the protective UPR appears to be one mechanism by which FABP inhibition damages MM cells. We also observed decreased *XBP1* expression and decreased XBP1 pathway activation with FABPi. Based on studies demonstrating the IRE/

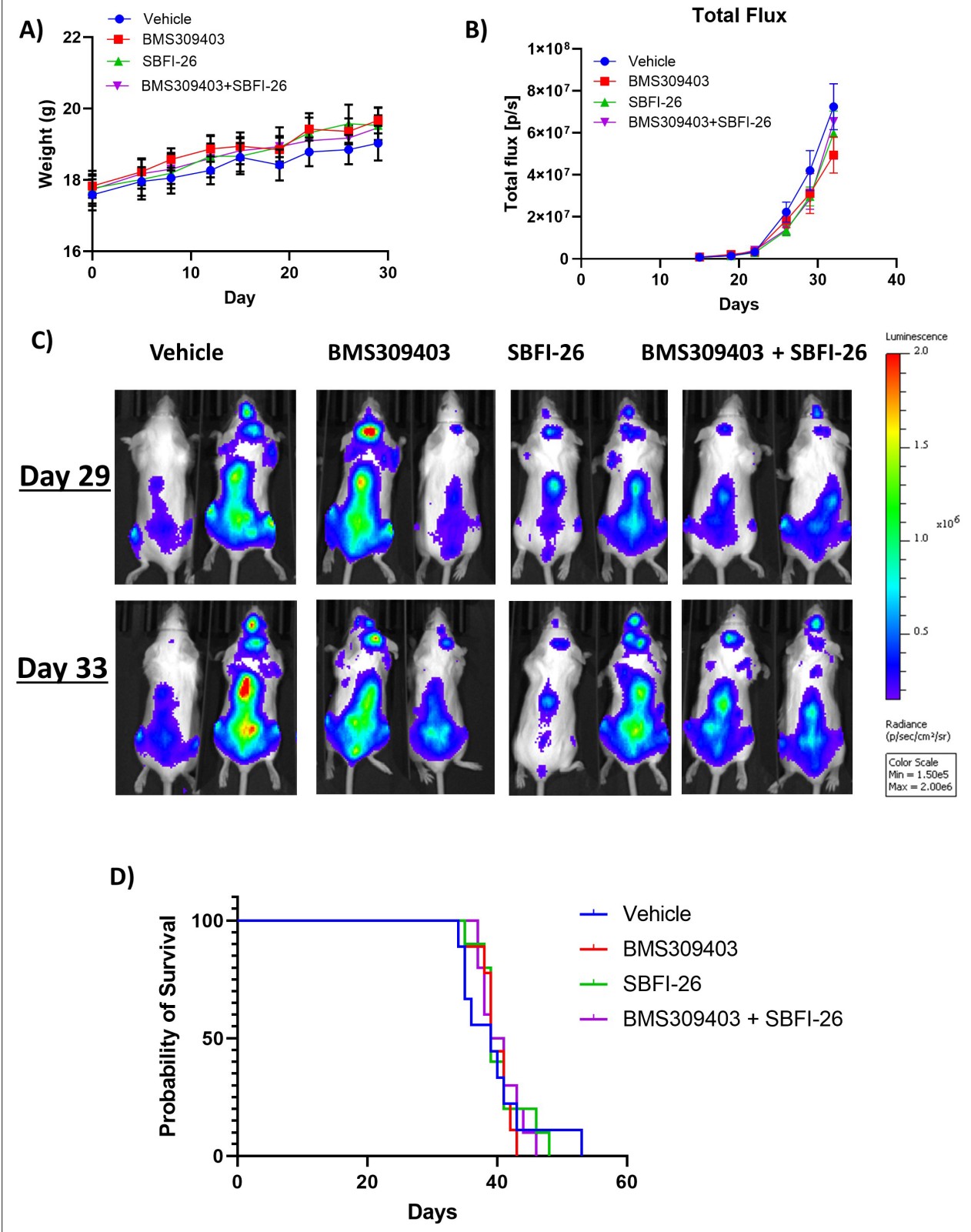

**Figure 7.** FABPi do not consistently increase survival or decrease tumor burden in myeloma xenograft mice (cohort 2). (**A**) Mouse weights from the second cohorts of SCID-beige- GFP+/Luc +MM.1 S mice treated with BMS309403, SBFI-26, or the combination from day of injection plotted as Mean ± SEM. (**B**) Tumor burden from two separate cohorts of SCID-beige GFP+/Luc +MM.1 S assessed by bioluminescence imaging (BLI) in MM.1S model. No significance detected with One-way ANOVA with Dunnett's multiple comparison test. (**C**) Representative BLI images from second cohort of SCID-

*Figure 7 continued on next page*

*Figure 7 continued*

Beige MM.1S[luc+] mice at days 29 and 33. (**D**) Survival of SCID-Beige GFP+/Luc +MM.1 S mice from second cohort- no significance observed. Analysis performed by Kaplan-Meier Survival Analysis, Log-Rank (Mantel-Cox) test, no significance in panel D, n=10.

XBP1 pathway is required for differentiation and survival of MM cells (*White-Gilbertson et al., 2013*), this could be a driver of the decreased UPR and MM cell death resulting from FABPi.

Interestingly, decreased UPR and XBP1 signaling could result from decreased MYC expression directly, since MYC directly controls IRE1 transcription by binding to its promoter and enhancer (*Zhao et al., 2018*). While others have shown that BMS309403 reduces UPR in skeletal muscle cells (*Bosquet et al., 2018*), this has not previously been shown in tumor cells before now. As a transcription factor, c-MYC can act as an activator or repressor through either direct binding to regulatory regions, or through chromatin modulation. A MYC activation signature is seen in 67% of MM patients (*Chng et al., 2011*), and this signature influences the progression from monoclonal gammopathy of undetermined significance (MGUS) to MM. Targeting MYC in MM cells by knockdown (*Cao et al., 2021*) or treatment with a small molecule inhibitor (*Holien et al., 2012*) induces cell death; however, the importance of MYC in many healthy cell types make targeting it difficult. Thus, our study represents a novel approach to reducing MYC by targeting the FABP family. This work also builds upon data that myeloma cells exhibit aberrant amino acid, lipid, and energy metabolism (*Steiner et al., 2018*), and data revealing the importance of metabolic enzymes in myeloma tumorigenesis (*Li et al., 2021*) and drug resistance (*Lipchick et al., 2021*) by demonstrating the role of FABPs in MM cell metabolism and mitochondrial integrity. In sum, we demonstrated that FABPs are a new protein family potentially important in MM.

Herein we demonstrated the pivotal role of FABPs in myeloma cell survival in vitro and in clinical datasets. However, in vivo results were mixed, and followup analysis needs to be performed before clinical work can be initiated, such as optimizing doses or delivery mechanisms and determing if any effects in vivo were due to the early drug administration (which could affect homing). More systemic analysis of mice, such as testing immune cell effects that could reduced efficacy of FABPi, is also needed since FABPi alter a plethora of phenotypes across the body, including glucose metabolism, lipid metabolism, and inflammation (*Bosquet et al., 2018*; *Lan et al., 2011*; *Shibue et al., 2015*) – all of which have potential implications for myeloma disease progression. Demonstration of efficacy of FABPi on established MM tumors in vivo, as well as effects of FABPi on primary MM cells, which we were not able to obtain in our laboratory, must also preceed clinical translation. Lastly, an assessment of the FABPi effects on tumor cells in vivo (e.g. effects on proliferation markers (proliferating cell nuclear antigen or Ki67) or apoptosis) would be reveal in vivo effects of FABPi.

## Conclusion

Pharmacologic or genetic inhibition of FABPs result in reduced growth, decreased UPR and MYC signaling, decreased metabolism, and induction of apoptosis in myeloma cells in vitro. FABP inhibition in vivo had variable effects. Patients that have high *FABP5* expression within their myeloma cells have worse outcomes and high *FABP5* is seen in MM clinical subtypes that have a more aggressive phenotype. Collectively, these data demonstrate the anti-myeloma effects of FABP inhibition, suggest different mechanisms driving this, and thus describe a potentially new target for MM therapy.

## Materials and methods

### Materials and reagents

Recombinant FABP4 (10009549) and FABP5 (10010364) were purchased from Caymen Chemical (Ann Arbor, MI). Dexamethasone (dex) (VWR), BMS3094013 (Caymen Chemical), SBFI-26 (Aobious, Gloucester, MA), and the MYC inhibitor 10058-F4 (Abcam, Cambridge, UK) were dissolved in DMSO. In vitro, dex was used at 80 µM; BMS309403 and SBFI-26 were used at 50 µM either as single treatments or in combination, unless otherwise stated.

### Cell culture

Human myeloma cell lines GFP+/Luc+MM.1S, MM.1S (ATCC, Manassas, VA), RPMI-8226 (ATCC), MM.1R (ATCC), OPM2 (DSMZ), and mouse cell line GFP+/Luc+ 5TGM1-TK (5TGM1-TK) were

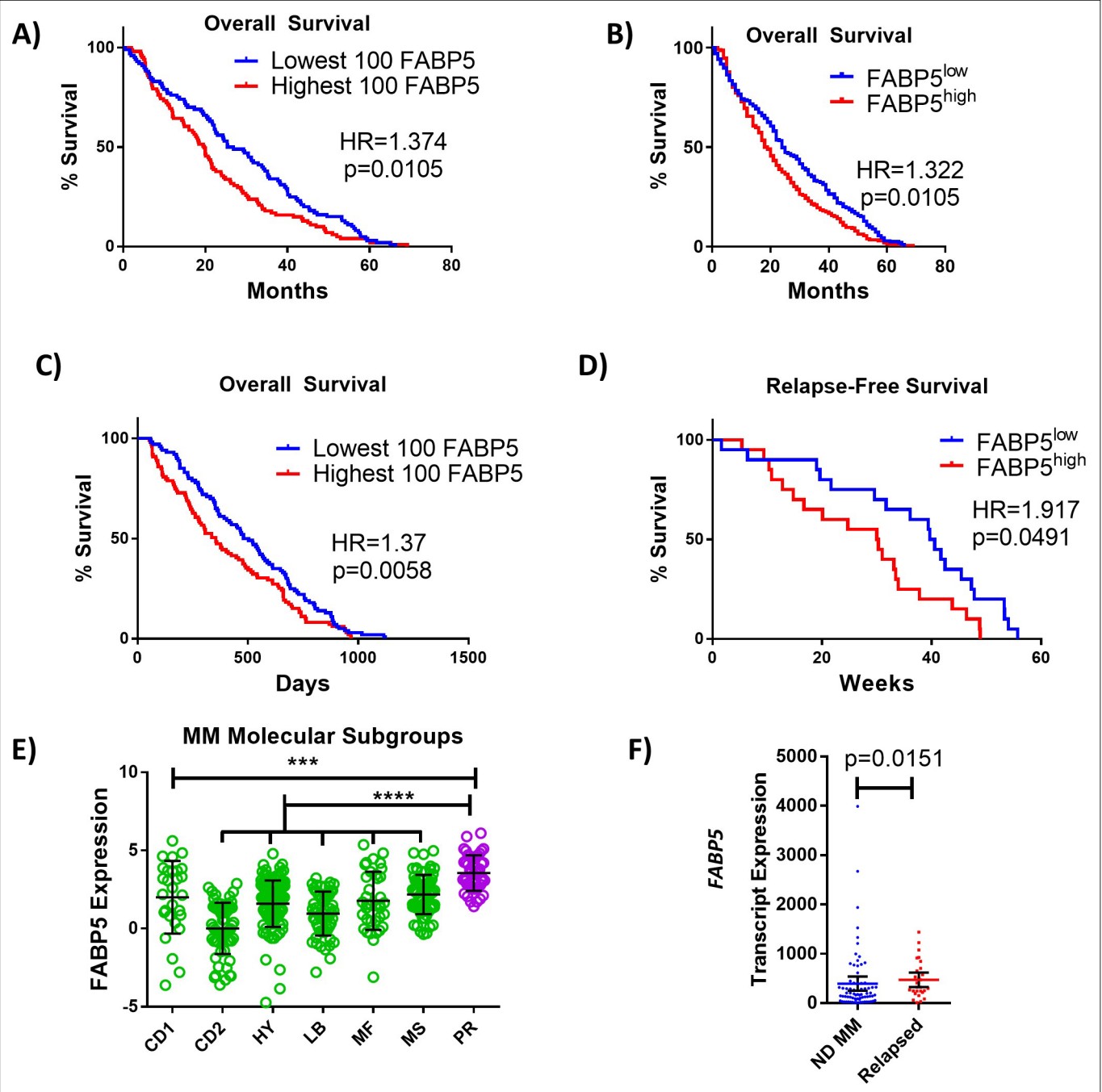

**Figure 8.** FABP proteins are clinically relevant in MM. (**A, B**) Kaplan-Meier analysis of overall survival (OS) of MM patients in Zhan et al. dataset stratified as top (n=100) or bottom (n=100) *FABP5* expressing, or all patients above (n=207) or below (n=207) the median. (**C**) Kaplan–Meier analysis of relapse-free survival of MM patient groups in Mulligan et al. dataset stratified as top (n=100) or bottom (n=100) *FABP5* expressing. (**D**) Kaplan–Meier analysis of relapse-free survival of MM patient groups in Carrasco et al. dataset: high (n=20) and low (n=20) *FABP5* relative to median. (**E**) Molecular subtypes of MM cells were analyzed for FABP5 expression and significance between all groups and the highly aggressive subtype (PR) was observed using a one-way ANOVA with Dunnett's multiple comparison testing. (CD1 or CD2 of cyclin D translocation; HY: hyperdiploid; LB: low bone disease; MF or MS with activation of MAF, MAFB, or FGRF3/MMSET; PR: proliferation. From reference *Zhan et al., 2006*). (**F**) Data from Chng et al. dataset from newly-diagnosed (ND) (n=73) and relapsed MM patients (n=28) as mean with 95% confidence interval (CI), with statistical analysis performed using a Mann Whitney test. Data are mean ± SD unless otherwise stated. *p<0.05. **p<0.01. ***p<0.001. ****p<0.0001. Please see 2 supplements to Figure 8.

The online version of this article includes the following figure supplement(s) for figure 8:

**Figure supplement 1.** CoMMpass Dataset analysis of FABP5 demonstrates decreased progression-free survival and overall-survival in MM patients with high FABP5 expression in MM cells.

*Figure 8 continued on next page*

**Figure supplement 2.** CoMMpass Dataset analysis of FABP6, 3, and 4 demonstrates decreased survival and overall-survival in MM patients with high FABP6 expression in MM cells.

maintained in standard MM cell media: RPMI-1640 medium, 10% FBS (Atlanta Biologicals, Flowery Branch, GA), and 1 X Antibiotic-Antimycotic (100 U/ml penicillin, 100 μg/ml streptomycin, 0.25 μg/ml fungizone) (ThermoFisher Scientific, Grand Island, NY). U266 (ATCC) cells were maintained in MM growth medium +15% FBS (Atlanta Biologicals). NCI-H929 (H929, ATCC) cells were maintained in MM growth medium plus 0.05 mM 2-mercaptoethanol. Vk*MYC cells were maintained in RPMI-1640 medium +20% FBS. Vk*MYC, and GFP$^+$/Luc$^+$MM.1S cells were generously provided by Dr. Ghobrial (Dana-Farber Cancer Institute). GFP$^+$/Luc$^+$ 5TGM1-TK cells were generously provided by Dr. Roodman (Indiana University). *FABP5* WT and KO MM.1R (ATCC) cells were generated by Synthego (Menlo Park, CA). Primary human MSCs were isolated from deidentified cancellous bone from the acetabulum received from donors (men and women) after total hip arthroplasty through the MaineHealth Biobank after IRB approval and informed consent (Biobank IRB # 2526). Human MSCs were isolated by surface adherence and cultured with a growth media of DMEM, 10% FBS, and 1% an antibiotic-antimycotic as previously described (*Fairfield et al., 2018*; *Reagan et al., 2014*; *Schutze et al., 2005*).

## Cell number quantification, cell cycle, and apoptosis in vitro assays

Cell numbers were measured by bioluminescence imaging (BLI), CellTiter Glo (Promega, Madison, WI), or RealTime Glo (Promega) assays, according to the manufacturer's instructions, and read on a GLOMAX microplate reader (Promega). Cell cycle analysis was measured with DAPI (0.5 μg/ml) and Ki67 staining (Alexa Fluor 647 Ki67 antibody, 350510, BioLegend). Apoptosis was measured using an annexin V/APC and DAPI Kit (BioLegend); total apoptotic cells were defined as annexin V$^+$/DAPI$^+$+annexin V$^+$/DAPI$^-$ populations. Data were acquired on a Miltenyi MACSquant flow cytometer and data analysis was performed using FlowJo software (BD Life Sciences). For BLI in vitro imaging of luciferase expressing cells, sterile luciferin (10 μL/well from a 7.5 mg/mL stock, VivoGlo, Promega) is added to white, 96 well plates of cells, given 5 min to reach equilibrium, and read in a GLOMAX microplate reader (Promega). For flow cytometry, a minimum of 10,000 events was collected and gated off forward and side scatter plots.

## Immunofluorescence and confocal microscopy

Myeloma cells were fixed and permeabilized using the Nuclear Factor Fixation and Permeabilization Buffer Set (Biolegend, San Diego CA), stained with DAPI (20 μg/ml), antibodies against FABP5 (MA5-2402911215, 1.25 μg/mL, ThermoFisher), and Alexa Fluor 647 anti-rabbit secondary antibody (A-21244, 1.25 μg/mL, ThermoFisher). Cells were then rinsed twice with PBS and imaged on a Leica SP5X laser scanning confocal microscope (Leica Microsystems, Buffalo Grove, IL) with Leica LAS acquisition software, using settings as previously described (*Fairfield et al., 2021*) using a 20×dry objective on 1.5 mm glass-bottomed dishes (MatTek Corporation, Ashland, MA).

## CRISPR/Cas9 FABP5-knockout MM.1R cell line development and characterization

An FABP5-KO pool of MM.1R cells and controls were generated by Synthego using the Guide target ACTTAACATTCTACAGGAGT, Guide sequence ACUUAACAUUCUACAGGAGU and PAM recognition sequence GGG. MM.1R were used as they were found to be the most amenable to CRISPR-Cas9 genetic targeting technology. MM.1R cells were obtained from ATCC by Synthego and confirmed as mycoplasma-negative and free from microbial contamination. Control and KO cell pools were provided to the Reagan lab at passage 4 and passage 5, respectively. Single-cell clones were not able to be expanded and thus the pooled sample was used. PCR and sequencing primers used for confirmation were: Fwd: TTTCATATATGTAAAGTGCTGGCTC and Rev:TGATACAGCCTATCATTCTAGAAGCT.

Wild type and edited cells were thawed and allowed to grow for 1 week prior to seeding (5000 cells/well; 96-well plate with Real Time Glo (RTG)). Cells from both pools were seeded at ~1 million cells/

T25 for 96 hr prior to harvest for RNA (Qiazol). The expression of FABP family members in both experiments was assessed by qRT-PCR.

## Western blotting
Protein from cell lysates was extracted using RIPA buffer (Santa Cruz, 24948) or Minute Total Protein Extraction Kit (Invent Biotechnology, SD-001/SN-002) and quantified using a DC protein assay kit II (Bio-Rad, 5000112). Samples were denatured in 4 x laemmli buffer (Bio-Rad, 1610747) with β-mercaptoethanol (VWR, 97064–880) for 5 min at 95 °C, run on 12% polyacrylamide gels (Bio-Rad, 5671043), and transferred onto PVDF membranes (Bio-Rad, 1704156). Blots were blocked for 2 hr in 5% non-fat milk (VWR, 10128–602). Staining protocols with antibody details are in *Supplementary file 23*. All antibodies were incubated at 4 °C. Blots were imaged after adding ECL reagents (Biorad, 1705060) for 5 min and visualized using Azure c600 (Azure biosystems).

## Seahorse metabolic assays
GFP+/Luc +MM.1 S cells were cultured for 24 hr with BMS309403 (50 µM), SBFI-26 (50 µM), or both and then adhered to Cell Tak (Corning)-coated Seahorse XF96 V3 PS cell culture microplates (Agilent, #101085–004) at a density of 60,000 cells/well in XF DMEM medium pH, 7.4 (Aglient #103576–100) supplemented with 1 mM sodium pyruvate, 2 mM glutamine and 10 mM glucose according to the manufacturer's instructions (https://www.agilent.com/cs/library/technicaloverviews/public/5991-7153EN.pdf). Oxygen consumption rate in cells was then measured in basal conditions and in response to oligomycin (1.25 µM), FCCP (1 µM), and rotenone and antimycin A (0.5 µM). Data were analyzed using Wave Software V2.6 and Seahorse XF Cell Mito Stress Test Report Generators (https://www.agilent.com). A one-way ANOVA was used for each parameter with Uncorrected Fisher's LSD multiple comparison post-hoc testing for significance. Results represent 5 independent experiments with 1 representative experiment shown with 20–24 wells per condition. In a separate set of experiments, cells were treated as above, however etomoxir or vehicle was added at a final concentration of 4 µM prior to subjecting the cells to the mitochondrial stress test. Due to artificial increases in OCR caused by further warming of the plate during ETOX measurements, the ETOX response data was normalized to MM.1S (vehicle, vehicle) control cells.

## TMRE mitochondrial membrane potential Assay
GFP[+]/Luc[+] MM.1S cells were cultured for 24, 48, and 72 hr with BMS309403 (50 µM), SBFI-26 (50 µM), or combination before staining with 0.5 mM TMRE for 30 minutes per Caymen Chemical protocol. Data acquisition was performed on a Miltenyi MACSquant flow cytometer and data analysis was performed using FlowJo analysis software (BD Life Sciences) with a minimum of 10,000 events collected and gated off forward and side scatter plots.

## CellROX green oxidative stress and MitoSOX red mitochondrial superoxide assays
ATCC MM.1S cells were cultured for 24, 48, and 72 hr with BMS309403 (50 µM), SBFI-26 (50 µM), or combination before staining with 500 nM CellROX for 30 min or 5 µM MitoSOX for 10 min per Thermofisher Scientific protocol. Data acquisition was performed on a Miltenyi MACSquant flow cytometer and data analysis was performed using FlowJo analysis software (BD Life Sciences) with a minimum of 10,000 events collected and gated off forward and side scatter plots.

## Quantification of global 5-hydroxymethylcytosine levels
DNA was isolated from 1 million GFP+/Luc +MM.1 S cells after 24 hr of treatment with vehicle (DMSO) or 50 µM BMS309403 and 50 µM SBFI-26 using the DNeasy Blood and Tissue kit (Qiagen, Germantown, MD, USA) per the manufacturer's instructions. DNA was quantified and tested for quality and contamination using a Nanodrop 2000 (Thermo Fisher Scientific) and subjected to quality control minimum standards of 260/230>2 and 260/280>1.8 prior to use in subsequent steps. 100 ng of DNA was then analyzed via MethylFlash Global DNA Hydroxymethylation (5-hmC) ELISA Easy Kit (Cat.# P-1032–48, Epigentek, Farmingdale, NY, USA) per the manufacturer's instructions.

## Quantitative RT-PCR

GFP+/Luc +MM.1 S, MM.1S, 5TGM1-TK, OPM-2 and RPMI-8226 cells were cultured for 24 hr with treatments prior to mRNA isolation as described above. cDNA synthesis (Applied Biosciences High Capacity cDNA Kit, ThermoScientific, Waltham, MA, USA) was executed prior to quantitative PCR (qRT-PCR) using SYBR Master Mix (Bio-Rad, Hercules, CA, USA) and thermocycling reactions were completed using a CFX-96 (Bio-Rad Laboratories). Data were analyzed using Bio-Rad CFX Manager 3.1 and Excel (Microsoft Corp., Redmond, WA, USA) using the delta-delta CT method. Primer details are in *Supplementary file 24*. Two wells (technical duplicates) were used at the minimum, for qRT-PCR analysis for each biological data point.

## Mass spectrometry proteomics

### Sample preparation

1. Cells for proteomics analysis were harvested by scraping into centrifuge tubes and pelleting for 5 min at 2500×g, 4 °C. Cells were then resuspended in PBS and pelleted, twice for a total of two cell pellet washes.
2. Cells were solubilized in ice-cold RIPA buffer and DNA sheared using a probe-tip sonicator (3×10 s) operating at 50% power with the samples on ice. Each was then centrifuged (14,000×g) at 4 °C and the supernatant collected. Protein content was measured relative to bovine serum albumin protein concentration standards using the bicinchoninic acid (BCA) assay (Thermo Scientific Pierce, Waltham, MA).
3. Approximately 100 µg protein from each sample was used in further sample preparation. Protein precipitation was initiated with the addition of a 10-fold volumetric excess of ice-cold ethanol. Samples were then placed in an aluminum block at –20 °C for 1 hr, then protein pelleted in a refrigerated tabletop centrifuge (4 °C) for 20 min at 16,000×g. The overlay was removed and discarded. Protein samples were allowed to dry under ambient conditions.
4. Each sample was resuspended in 50 mM Tris (pH = 8.0) containing 8.0 M urea and 10 mM TCEP (tris(2-carboxyethyl)phosphine hydrochloride, Strem Chemicals, Newburyport, MA). Reduction of cysteine residues was performed in an aluminum heating block at 55 °C for 1 hr.
5. After cooling to room temperature, each sample was brought to 25 mM iodoacetamide (Thermo Scientific Pierce, Waltham, MA) and cysteine alkylation allowed to proceed for 30 min in the dark. Reactions were quenched with the addition of 1–2 µL 2-mercaptoethanol (Thermo Scientific, Waltham, MA) to each sample.
6. Each was diluted with 50 mM Tris buffer (pH = 8.0–8.5) containing 1.0 mM calcium chloride (Sigma-Aldrich, St. Louis MO) such that the urea concentration was brought below 1.0 M. Sequencing-grade modified trypsin (Promega, Madison, WI) was added to a final proportion of 2% by mass relative to sample total protein as measured with the BCA assay. Proteolysis was performed overnight at 37 °C in the dark.
7. Samples were evaporated to dryness using a centrifugal vacuum concentrator. Each was redissolved in 4% acetonitrile solution containing 5% formic acid (Optima grade, Fisher Scientific, Waltham, MA). Peptides were freed of salts and buffers using Top Tip Micro-spin columns packed with C18 media (Glygen Corporation, Columbia, MD) according to manufacturer-suggested protocol.
8. Samples were again evaporated to dryness using a centrifugal vacuum concentrator and peptides redissolved in 4% acetonitrile solution containing 5% formic acid (Optima grade).

## LC-MS/MS

All sample separations performed in tandem with mass spectrometric analysis are performed on an Eksigent NanoLC 425 nano-UPLC System (Sciex, Framingham, MA) in direct-injection mode with a 3 µL sample loop. Fractionation is performed on a reverse-phase nano HPLC column (Acclaim PepMap 100 C18, 75 µm×150 mm, 3 µm particle, 120 Å pore) held at 45 °C with a flow rate of 350 nL/min. Solvents are blended from LC-MS-grade water and acetonitrile (Honeywell Burdick & Jackson, Muskegon, MI). Mobile phase A is 2% acetonitrile solution, while mobile phase B is 99.9% acetonitrile. Both contain 0.1% formic acid (Optima grade, Fisher Chemical, Waltham, MA). Approximately 1 µg of peptides are applied to the column equilibrated at 3% B and loading continued for 12 min. The sample loop is then taken out of the flow path and the column washed for 30 s at starting conditions. A gradient to 35% B is executed at constant flow rate over 90 min followed by a 3 min gradient to

90% B. The column is washed for 5 min under these conditions before being returned to starting conditions over 2 min.

Analysis is performed in positive mode on a TripleTOF 6600 quadrupole time-of-flight (QTOF) mass spectrometer (Sciex, Framingham, MA). The column eluate is directed to a silica capillary emitter (SilicaTip, 20 µm ID, 10 µm tip ID, New Objective, Littleton, MA) maintained at 2400–2600 V. Nitrogen nebulizer gas is held at 4–6 psi, with the curtain gas at 21–25 psi. The source is kept at 150 °C.

Data acquisition performed by information-dependent analysis (IDA) is executed under the following conditions: a parent ion scan is acquired over a range of 400–1500 mass units using a 200 ms accumulation time. This is followed by MS/MS scans of the 50 most-intense ions detected in the parent scan over ranges from 100 to 1500 mass units. These ions must also meet criteria of a $2^+$–$5^+$ charge state and of having intensities greater than a 350 counts-per-second (cps) threshold to be selected for MS/MS. Accumulation times for the MS/MS scans are 15 ms. Rolling collision energies are used according to the equation recommended by the manufacturer. Collision energy spread is not used. After an ion is detected and fragmented, its mass is excluded from subsequent analysis for 15 s.

SWATH analysis is performed according to previously-published optimized conditions tailored to the 6600 instrument (*Schilling et al., 2017*). Briefly, SWATH MS/MS windows of variable sizes are generated using Sciex-provided calculators. Rolling collision energies are used, as well as fragmentation conditions optimized for ions of a $2^+$ charge state. SWATH detection parameters are set to a mass range of m/z=100–1500 with accumulation times of 25 ms in the high-sensitivity mode. A parent-ion scan is acquired over a range of 400–1500 mass units using a 250 ms accumulation time. The PRIDE (PRoteomics IDEtifications Database) was used to upload and share raw data (*Perez-Riverol et al., 2019*), and InteractiveVenn software was used to make Venn Diagrams to combine Proteomic and RNAseq data (http://www.interactivenn.net/#) (*Heberle et al., 2015*). Heatmaps of proteomic data were generated using centroid linkage and Kendall's Tau distance measurement algorithms with http://www.heatmapper.ca/expression.

## Cell line validation

Cells were authenticated and validated as mycoplasma and virus negative by the Yale Comparative Pathology Research Core on the following dates.

| Cell Line | Source | Cell Authentication | Mycoplasma Test | Number of Passages |
|---|---|---|---|---|
| GFP$^+$/Luc$^+$ MM.1S | Ghobrial Laboratory, 2015 | STR panel, UVM (University of Vermont), 2022 | 2022 | 1–30 |
| MM.1S | ATCC | STR panel, University of Vermont, 2022 | 2022 | 1–30 |
| GFP$^+$/Luc$^+$ 5TGM1-TK | Roodman Laboratory, 2015 | Not possible at this time | 2021 | 1–30 |
| RPMI-8226 | ATCC | STR panel, University of Vermont, 2022 | 2016 | 1–30 |
| OPM-2 | DSMZ | STR panel, University of Vermont, 2022 | 2022 | 1–30 |
| Vk*Myc mouse cells | Ghobrial Laboratory, 2021 | Not possible at this time | 2016 | 1–30 |
| MM.1R | ATCC | STR panel, University of Vermont, 2022 | 2022 | 1–30 |
| U266 | ATCC | STR panel, University of Vermont, 2022 | 2022 | 1–30 |
| NCI-H929 | ATCC | N/A | 2021 | 1–30 |

## In vivo experiments

All experimental studies and procedures involving mice were performed in accordance with approved protocols from the MaineHealth Institute for Research (Scarborough, Maine, USA) Institutional Animal Care and Use Committee (#1812 and 2111). In cohort one, eight week old female SCID-beige (CB17. Cg-PrkdcscidLystbg-J/Crl, Charles River) mice were inoculated intravenously (IV) with 5x10^6 GFP$^+$/Luc$^+$MM.1S cells by a blinded investigator. Mice were randomized based on weight and body parameters, then treatments then began 3 X/week with either 5 mg/kg BMS309403, 1 mg/kg SBFI-26, the combination (5 mg/kg BMS309403 +1 mg/kg SBFI-26), or the vehicle (5% DMSO), intraperitoneally

(n=12/group), based on safe doses reported previously (*Al-Jameel et al., 2017*; *Yan et al., 2018*). Body parameters were assessed with piximus at day 1 and 30. In a second cohort of SCID-Beige mice (n=10/group, randomized by weight), a near identical experimental schema was followed, except body parameters were not assessed. In a second animal model, 10–12 week old mice (both sexes, mixed equally between groups) of KaLwRij/C57Bl6 mice (from Dana-Farber Cancer Institute) were injected with 1x10^6 GFP+/Luc+ 5TGM1-TK cells IV by a blinded investigator, randomized by weight, and treated as in the SCID-Beige model with 5 mg/kg BMS309403 (n=9) or vehicle (n=8). Mice were frequently weighed and monitored for clinical signs of treatment-related side effects. "Survival endpoints" were mouse death or euthanasia as required by IACUC, based on body conditioning score including weight loss and impaired hind limb use. Survival differences were analyzed by Kaplan-Meier methodology. For bioluminescent imaging, mice were injected with 150 mg/kg i.p. filter-sterilized D-luciferin substrate (VivoGlo, Promega) and imaged after 15 min in an IVIS Lumina LT (Perkin Elmer, Inc; Waltham, MA). Tail vein injector was blinded in all studies; BLI technician was blinded in second SCID-Beige study. Data were acquired and analyzed using LivingImage software 4.5.1. (PerkinElmer). Body parameters (BMD, BMC, Lean Mass, and Fat Mass) were measured with PIXImus duel-energy X-ray densitometer (GE Lunar, Boston, MA, USA). The PIXImus was calibrated daily with a mouse phantom provided by the manufacturer. Mice were anesthetized using 2% isoflurane via a nose cone and placed ventral side down with each limb and tail positioned away from the body. Full-body scans were obtained and DXA data were gathered and processed (Lunar PIXImus 2, version 2.1). BMD and BMC were calculated by extrapolating from a rectangular region of interest (ROI) drawn around one femur of each mouse, using the same ROI for every mouse, and lean and fat mass were also calculated for the entire mouse, exclusive of the head, using Lunar PIXImus 2.1 software default settings. Each mouse (single animals) was considered the experimental unit (rather than litters or cage of animals). Replicates numbers were decided from experience of the techniques performed and practical considerations. Mice that didn't have reliable *IV* injections were noted to be dropped, as agreed upon a priori. To minimize confounders, cages were chosen at random for IV tumor injections, and needles loaded with tumor cells were pre-loaded and laid out and then chosen at random. The ARRIVE guidelines (Animal Research: Reporting of In Vivo Experiments), a checklist of information to include in publications describing animal research, was followed.

## mRNA isolation and RNA-Seq

Three biological sets of GFP+/Luc+MM.1S cells were cultured for 24 hr with vehicle, 50 µM BMS309403, 50 µM SBFI-26, or the combination prior to mRNA isolation with Qiazol (Qiagen, Germantown, MD) and miRNeasy Mini Kit with on-column DNAse digestion (Qiagen) according to the manufacturer's protocol. Samples underwent library preparation, sequencing, and analysis at the Vermont Integrative Genomics Resource. mRNA was quantified and tested for quality and contamination using a Nanodrop (Thermo Fisher Scientific) and subjected to quality control standards of 260/230>2 and 260/280>1.8 prior to library preparation. Partek Flow (version 10.0.21.0302) was used to analyze the sequence reads. Poorer quality bases from the 3' end were trimmed (phred score <20), and the trimmed reads (ave. quality >36.7, ave. length 75 bp, ave. GC ~56%) were aligned to the human reference genome hg38 using the STAR 2.6 aligner. Aligned reads were then quantified using an Expectation-Maximization model, and translated to genes. Genes that had fewer than 30 counts were then filtered, retaining 14,089 high count genes. Differentially expression comparisons were performed using DESeq2. Downstream comparisons of IPA canonical pathways and upstream regulators were executed in Excel (Microsoft, Redmond, WA). Data were analyzed through the use of IPA2 (QIAGEN, https://www.qiagenbioinformatics.com/products/ingenuitypathway-analysis) and STRING DB version 11.0. RNAseq heatmap of Myc pathway was generated on http://www.heatmapper.ca/expression applying clustering to rows and columns using single linkage and Pearson distance measurement algorithms.

## Cancer dependency map (DepMap) analysis

Genetic dependency data from the Dependency Map (DepMap) Portal's CRISPR (Avana) Public20Q3 (https://depmap.org/portal/download/) of 20 human MM cell lines were analyzed and the dependency score (computational correction of copy-number effect in CRISPR-Cas9 essentiality screens

(CERES)) of Hallmark Fatty Acid Metabolism genes from Gene Set Enrichment Analysis (https://www.gseamsigdb.org) were determined.

## Survival and expression analyses of clinical datasets

The (*Zhan et al., 2006*) (GSE132604), (*Carrasco et al., 2006*) (GSE4452), and (*Mulligan et al., 2007*) (GSE9782) datasets were analyzed using OncoMine (ThermoFisher). The Chng dataset (*Chng et al., 2007*) showing patient *FABP5* mRNA transcript data was analyzed from accession number GEO:GSE6477. The relationship between *FABP5* and MM progression was analyzed with Kaplan-Meier analysis using log-rank Hazard Ratio (HR) and Gehan-Breslow-Wilcoxon significance testing. Gene expression data were downloaded (GEO; GSE6477), log-transformed, and analyzed with an one-way ANOVA model using the aov() function in R, as previously described (*Fairfield et al., 2021*).

For survival analysis in the CoMMpass dataset, survival and Transcripts Per Million (TPM)-normalized gene expression data (IA15 data release) were downloaded from the Multiple Myeloma Research Foundation (MMRF)'s Researcher Gateway (6/16/2021). Patient samples drawn at timepoints other than the baseline were removed from consideration. Based on the histogram of FABP5 expression levels in the CoMMpass cohort, FABP5 expression follows a right-tailed distribution, whereby a subset of patient tumors exhibit higher levels of FABP5. We discretized FABP5 expression based on the cohort's mean (10.838), stratified samples as FABP5-high and FABP5-low and plotted Kaplan-Meier curves to showcase its effect on OS and PFS. To derive effect estimates, we examined associations between FABP5-high (vs. FABP5-low) in a Cox proportional Hazards Model. Exploratory general linear models also examined the association between BMI and FABP5 expression levels, adjusting for age and sex. Based on the boxplot generated to identify related FABP gene expression levels, FABP3, FABP4 and FABP6 were also significantly expressed in myeloma cells. Thus, following similar procedures, analyses were also conducted based on the cohort's mean for FABP3 (3.2611), FABP4 (1.624), and FABP6 (0.786).

## Statistical analysis

Data were analyzed using GraphPad Prism v.6 or above, and unpaired Student's t tests or one-way or two-way ANOVA using Tukey's correction was performed, unless otherwise stated. Data are expressed as mean ± standard error of the mean (SEM) or standard deviation (SD); ****$p \leq 0.0001$; ***$p < 0.001$; **$p < 0.01$; *$p < 0.05$.

## Acknowledgements

We thank Dr. Christine Lary for downloading and processing the GSE6477 dataset and Lauren Lever, Samantha Costa, Dr. Clifford Rosen, Madeleine Nowak, Dr. Matt Lynes, and the Vermont Integrative Genomics Resource staff for intellectual or technical contributions. The content is solely the responsibility of the authors and does not necessarily represent the official views of the NIH.

## Additional information

### Competing interests

Catherine R Marinac: GRAIL Inc: Research Funding; JBF Legal: Consultancy. Michaela R Reagan: Reviewing editor, eLife. The other authors declare that no competing interests exist.

### Funding

| Funder | Grant reference number | Author |
| --- | --- | --- |
| National Cancer Institute | F31CA257695 | Connor S Murphy |
| National Cancer Institute | R37CA245330 | Michaela R Reagan |
| National Cancer Institute | R50CA265331 | Heather Fairfield |
| National Institute of General Medical Sciences | P20GM103449 | Julie A Dragon |

| Funder | Grant reference number | Author |
| --- | --- | --- |
| National Institute of General Medical Sciences | U54GM115516 | Michaela R Reagan Heather Fairfield |
| National Institute of General Medical Sciences | P20GM121301 | Calvin Vary |
| American Cancer Society | RSG-19-037-01-LIB | Michaela R Reagan |
| American Cancer Society | IRG-16-191-33 | Michaela R Reagan |

The funders had no role in study design, data collection and interpretation, or the decision to submit the work for publication.

## Author contributions

Mariah Farrell, Heather Fairfield, Conceptualization, Data curation, Formal analysis, Validation, Investigation, Visualization, Methodology, Writing – original draft, Project administration, Writing – review and editing; Michelle Karam, Anastasia D'Amico, Investigation; Connor S Murphy, Formal analysis, Investigation, Visualization, Writing – review and editing; Carolyne Falank, Conceptualization, Investigation, Writing – review and editing; Romanos Sklavenitis Pistofidi, Investigation, Visualization, Methodology, Writing – review and editing; Amanda Cao, Investigation, Visualization, Methodology; Catherine R Marinac, Formal analysis, Investigation, Visualization, Methodology, Writing – review and editing; Julie A Dragon, Data curation, Formal analysis, Visualization; Lauren McGuinness, Reagan Di Iorio, Investigation, Writing – review and editing; Carlos G Gartner, Data curation, Investigation, Writing – review and editing; Edward Jachimowicz, Software, Formal analysis, Investigation, Writing – review and editing; Victoria DeMambro, Data curation, Software, Formal analysis, Investigation, Writing – review and editing; Calvin Vary, Data curation, Formal analysis, Investigation, Writing – review and editing; Michaela R Reagan, Conceptualization, Resources, Data curation, Formal analysis, Supervision, Funding acquisition, Validation, Investigation, Visualization, Methodology, Writing – original draft, Project administration, Writing – review and editing

## Author ORCIDs

Heather Fairfield http://orcid.org/0000-0002-8852-2254
Michaela R Reagan http://orcid.org/0000-0003-2884-6481

## Ethics

Human subjects: Primary human MSCs were isolated from deidentified cancellous bone from the acetabulum received from donors (men and women) after total hip arthroplasty through the Maine-Health Biobank after IRB approval and informed consent (Biobank IRB # 2526).

All experimental studies and procedures involving mice were performed in accordance with approved protocols from the Maine Medical Center Research Institute's (Scarborough, Maine, USA) Institutional Animal Care and Use Committee (protocols #2111 or #1812).

## Decision letter and Author response

Decision letter https://doi.org/10.7554/eLife.81184.sa1
Author response https://doi.org/10.7554/eLife.81184.sa2

# Additional files

## Supplementary files

• Supplementary file 1. RNAseq values for FABP genes in MM.1S and RPMI8226 cells demonstrated that FABP5 (italic) is the most highly expressed.

• Supplementary file 2. IC50 of myeloma cell lines treated with various doses overtime with BMS309403. Data includes the µM and logµM of human and mouse myeloma cell lines at 1, 24, 48, 72 hour timepoints.

• Supplementary file 3. IC50 of myeloma cell lines treated with various doses overtime with SBFI-26. Data includes the µM and log µM of human and mouse myeloma cell lines at 1, 24, 48, 72 hour timepoints.

• Supplementary file 4. List of genes changed by FABP inhibitors in MM.1S cells determined with

RNA sequencing. (A) BMS309403, SBFI-26, and the combination alter gene expression levels. DEGs were determined using a false discovery rate of 0.2.

• Supplementary file 5. List of canonical pathways changed by FABP inhibitors in MM.1S cells determined by Ingenuity Pathway Analysis (IPA). The fifteen pathways below were commonly dysregulated in all three treatment groups (BMS309403, SBFI-26, and the combination) compared to vehicle. Significance was determined in each treatment compared to vehicle using IPA default p-value cutoff ($P<0.05$) and lists of canonical pathways for each treatment group were cross-referenced with one another in Microsoft excel to generate this list.

• Supplementary file 6. List of upregulated pathways after FABP inhibitor treatment in MM.1S cells determined with RNA sequencing. (A) STRING revealed biological pathways upregulated after BMS309403, SBFI-26, and the combination treatment. Data filtered based on false discovery rate.

• Supplementary file 7. List of downregulated pathways after FABP inhibitor treatment in MM.1S cells determined by RNA sequencing. (A) STRING analysis revealed biological processes downregulated by combination treatment. Data filtered based on false discovery rate.

• Supplementary file 8. RNA sequencing reveals top 10 downregulated genes with BMS309403 and SBFI-26 treatment. Data filtered based on false discovery rate.

• Supplementary file 9. Protein expression changes due to 48 hour exposure to FABP inhibitors in MM.1S cells determined with mass spectrometry. (A) BMS309403, SBFI-26, and the combination alter gene expression levels. Data determined using |fold change|>1.2, and $P$-value ≥0.05.

• Supplementary file 10. List of 162 upregulated proteins after 48 hour exposure to BMS309403 in MM.1S cells determined with mass spectrometry proteomic analysis. Data determined using |fold change|>1.2, and $P$-value ≥0.05.

• Supplementary file 11. List of 51 upregulated proteins after 48 hour exposure to SBFI-26 in MM.1S cells determined with mass spectrometry proteomic analysis. Data determined using |fold change|>1.2, and $P$-value ≥0.05.

• Supplementary file 12. List of 199 upregulated proteins after 48 hour exposure to BMS309403 and SBFI-26 in MM.1S cells determined with mass spectrometry proteomic analysis. (A) BMS309403, SBFI-26, and the combination alter protein expression levels. Data determined using |fold change|>1.2, and $P$-value ≥0.05.

• Supplementary file 13. List of 177 downregulated proteins after 48 hour exposure to BMS309403 in MM.1S cells determined with mass spectrometry proteomic analysis. Data determined using |fold change|>1.2, and $P$-value ≥0.05.

• Supplementary file 14. List of 94 downregulated proteins after 48 hour exposure to SBFI-26 in MM.1S cells determined with mass spectrometry proteomic analysis. Data determined using |fold change|>1.2, and $P$-value ≥0.05.

• Supplementary file 15. List of 215 downregulated proteins after 48 hour exposure to BMS309403 and SBFI-26 in MM.1S cells determined with mass spectrometry proteomic analysis. Data determined using |fold change|>1.2, and $P$-value ≥0.05.

• Supplementary file 16. List of commonly upregulated proteins after 48 hour exposure to FABP inhibitors in MM.1S cells determined with mass spectrometry proteomics. Data determined using |fold change|>1.2, and $P$-value ≥0.05.

• Supplementary file 17. List of commonly downregulated proteins after 48 hour exposure to FABP inhibitors in MM.1S cells determined with mass spectrometry proteomics. Data determined using |fold change|>1.2, and $P$-value ≥0.05.

• Supplementary file 18. List of top 10 most significant upstream regulators predicted from IPA analysis of proteomic data in MM.1S cells exposed to BMS309403.

• Supplementary file 19. List of top 10 most significant upstream regulators predicted from IPA analysis of proteomic data in MM.1S cells exposed to SBFI-26.

• Supplementary file 20. List of top 10 most significant upstream regulators predicted from IPA analysis of proteomic data in MM.1S cells exposed to BMS309403 +SBFI-26.

• Supplementary file 21. 171 Molecules dysregulated by MYC with BMS309403 as identified by Ingenuity Pathway Analysis from RNAseq data.

• Supplementary file 22. Pearson Correlation Table of FABP5 correlation with other genes in the CoMMpass MMRF dataset. Only top 30 genes shown.

• Supplementary file 23. Antibody staining for western blotting. CST (Cell Signaling Technologies).

• Supplementary file 24. Primers used for reverse-transcriptase PCR. Forward (Fwd), Reverse (Rev); citation used if applicable.

• MDAR checklist

• Source data 1. Western blot raw data for MM1S stained for FABP5 or Myc, and 5TGM1-TK cells stained for Myc as labeled.

## Data availability

The clinical datasets used and analyzed during the current study are from Oncomine or data related to accession number GEO:GSE6477. RNA-seq data have been deposited in the NCBI Gene Expression Omnibus (GEO) database with the accession number GSE190699. The mass spectrometry proteomic data have been deposited to the ProteomeXchange Consortium via the PRIDE partner respository with the dataset identifier PXD032829.

The following datasets were generated:

| Author(s) | Year | Dataset title | Dataset URL | Database and Identifier |
|---|---|---|---|---|
| Farrell M, Fairfield H, Karam M, D'amico A, Murphy CS, Falank C, Pistofidis RS, Cao A, Marinac CR, Dragon J, McGuinness L, Gartner C, Iorio RD, Jachimowicz E, DeMambro V, Vary C, Reagan MR | 2022 | Fatty acid binding proteins contribute to multiple myeloma cell maintenance through regulation of Myc, the unfolded protein response, and metabolism | https://www.ncbi.nlm.nih.gov/geo/query/acc.cgi?acc=GSE190699 | NCBI Gene Expression Omnibus, GSE190699 |
| Farrell M, Fairfield H, Karam M, D'amico A, Murphy CS, Falank C, Pistofidis RS, Cao A, Marinac CR, Dragon J, McGuinness L, Gartner C, Iorio RD, Jachimowicz E, DeMambro V, Vary C, Reagan MR | 2023 | Fatty Acid Binding Protein 5 is a Novel Target in Multiple Myeloma | http://proteomecentral.proteomexchange.org/cgi/GetDataset?ID=PXD032829 | ProteomeXchange, PXD032829 |

The following previously published dataset was used:

| Author(s) | Year | Dataset title | Dataset URL | Database and Identifier |
|---|---|---|---|---|
| Chng WJ, Kumar S, Vanwier S, Ahmann G | 2007 | Expression data from different stages of plasma cell neoplasm | https://www.ncbi.nlm.nih.gov/search/all/?term=GSE6477 | NCBI Gene Expression Omnibus, GSE6477 |

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
