## [Editor Report]

This paper has valuable findings that have practical implications within multiple myeloma and tumor microenvironment fields. It describes a family of genes regulating myeloma cell survival and proliferation. The approaches used are convincing, state-of-the-art, and rigorous. The data support the essential claims of the manuscript and have mechanistic depth.

---

## [Decision Letter]

**Decision letter after peer review:**

Thank you for submitting your article "The Fatty Acid Binding Protein Family Represents a Novel Target in Multiple Myeloma" for consideration by *eLife*. Your article has been reviewed by 3 peer reviewers, one of whom is a member of our Board of Reviewing Editors, and the evaluation has been overseen by Mone Zaidi as the Senior Editor. The reviewers have opted to remain anonymous.

Essential revisions:

1) All three reviewers highlighted the lack of mechanistic detail as a key detractor of the manuscript. Please thoughtfully address those comments and include new findings as appropriate.

*Reviewer #1 (Recommendations for the authors):*

Overall the manuscript is very well written with excellent figures and supplemental data. Kudos to the authors for an excellent paper.

The major weakness is related to an unclear mechanism of action (MOA). While there is strong data suggesting that MYC is a critical component for FDBPs in MM, the authors also raise the possibility that targets such as EIF5B-a translation initiation factor that antagonizes cell cycle arrest via p21/p27-as well as other targets such as the IRE/XBP1 pathway may be crucial to the MOA. To strengthen the paper--similar to the experiments with dexamethasone--it would be interesting to see the use of inhibitors for other key components in the aforementioned MOA pathways also targeted via titration. For example, does using a MYC inhibitor at an efficacious dose abrogate any effects of FABP5 inhibition?

*Reviewer #2 (Recommendations for the authors):*

It is important to have a good understanding of FABP expression, beyond RNASeq data. This should be shown at gene and protein level for a panel of MM cell lines.

The mechanism is weak. While a detailed mechanistic study is beyond the scope of this study, a better understanding of FABP family expression both at baseline and in response to FABP inhibition is important.

*Reviewer #3 (Recommendations for the authors):*

1) Despite performing a very comprehensive analysis of the transcriptome, proteome, and metabolic profile of myeloma cells treated with FABPi, the authors did not identify the mechanisms(s) responsible for the decreased cell number. Studies investigating the role of MYC- or UPR-related genes in the response to FAPBi would increase the significance of the findings.

---

## [Author Response]

Reviewer #1 (Recommendations for the authors):Overall the manuscript is very well written with excellent figures and supplemental data. Kudos to the authors for an excellent paper.The major weakness is related to an unclear mechanism of action (MOA). While there is strong data suggesting that MYC is a critical component for FDBPs in MM, the authors also raise the possibility that targets such as EIF5B-a translation initiation factor that antagonizes cell cycle arrest via p21/p27-as well as other targets such as the IRE/XBP1 pathway may be crucial to the MOA. To strengthen the paper--similar to the experiments with dexamethasone--it would be interesting to see the use of inhibitors for other key components in the aforementioned MOA pathways also targeted via titration. For example, does using a MYC inhibitor at an efficacious dose abrogate any effects of FABP5 inhibition?

We thank the reviewer for this review with great insight in investigating the mechanisms of action. We have now completed the suggested experiment, a combination study of increasing doses of BMS309403 or SBFI-26 in the presence of a Myc inhibitor, 10058-F4. As now demonstrated in Figure 4E,F, FABP inhibitors are less effective in the presence of the Myc inhibitor. Thus, we have added a new Results section as follows:

“To test if MYC inhibition was a major cause of the FABPi effects on MM cells, we then pharmacologically inhibited MYC and tested a range of doses of FABPi. MYC inhibition alone dramatically reduced cell numbers at 72 hours, as expected, and FABP inhibition had less of an effect on MM cells when MYC was already inhibited (seen by a slope of ~0 for the black lines) (Figure 4E, F). This suggests that much of the effect of FABPi is through decreased MYC signaling, although the strong effect of the MYC inhibitor makes this difficult to determine unhesitantly. Similar results were seen at 24 and 48 hours (Figure 4—figure supplement 1).”

In addition, we investigated the role of reactive oxygen species (ROS) in FABP inhibition. We found that inhibition of FABPs lead to an increase in ROS, especially superoxide, which we have now added to the manuscript as Figure 5A,B and in Supplementary files.

Although not in our manuscript, we would like to share the data, see Author response image 1, where we recovered the FABP inhibitor-driven increase in ROS with N-acteyl cysteine (NAC) using doses described in the literature for this type of work. However, NAC also elicited a strong anti-myeloma effect on its own so the combination did not rescue the apoptosis that was induced with the FABP inhibitors (data not shown). More studies will need to be conducted to examine if there is an antioxidant that rescues the ROS induction without adding to the apoptotic phenotype of FABP inhibition. At this point, we have not been able to prove or conclude that FABPi-induced ROS drive any of the phenotypes we observed in MM cells, and thus we have not written that as a mechanism in our manuscript.

**Author response image 1. sa2fig1:** 

Reviewer #2 (Recommendations for the authors):It is important to have a good understanding of FABP expression, beyond RNASeq data. This should be shown at gene and protein level for a panel of MM cell lines.

We thank the reviewer for their question. To better understand FABP expression in myeloma cells and cells of similar lineage, we investigated both gene and protein expression levels of FABPs in the cancer cell line encyclopedia (CCLE). The Results section now includes our finding as follows “FABP5 consistently showed the expression in haematopoetic/lymphoid lineage lines within the Cancer Cell Line Encyclopedia (CCLE) at the gene level (Figure 1—figure supplement 1B) and protein level (Figure 1—figure supplement 1C) (“DepMap 22Q2 Public,” n.d.; Ghandi et al., 2019; Nusinow et al., 2020). In MM cell lines specifically, *FABP5* was the most highly expressed at the gene level (Figure 1B) and FABP5 and FABP6 were the most highly expressed at the protein level (Figure 1—figure supplement 1D).” We also included CoMMpass patient data in Figure 8—figure supplement 1 that indicates FABP3, 4, 5, and 6 are expressed within myeloma patient cells. All data highlight FABP5 as the highest expressed FABP family member.

The mechanism is weak. While a detailed mechanistic study is beyond the scope of this study, a better understanding of FABP family expression both at baseline and in response to FABP inhibition is important.

We thank the reviewer for their observation.

To better understand the expression of FABP family members at baseline, we utilized RNA-Seq and Mass Spectrometry proteomics from the Broad Cancer Cell Line Encyclopedia (CCLE) to investigate gene and protein expression of the FABP family members in a large swath of malignant cells of similar lineage. We found that *FABP5* was the predominant FABP family member expressed, followed by *FABP6* in the RNA-Seq gene expression data from 215 cell lines of hematopoietic and lymphoid lineage (Figure 1 figure supplement 1B), 30 of which were myeloma cell lines specifically (Figure 1B). In addition, FABP5 and FABP6 were also consistently the highest proteins expressed in the CCLE proteomics dataset in cells of hematopoietic/lymphoid lineage (Figure 1 figure supplement 1C) and MM cells specifically (Figure 1 figure supplement 1D). We also included CoMMpass patient data (Figure 8—figure supplement 1A) that indicate FABP3, 4, 5, and 6 are expressed within myeloma patient cells. All data highlight FABP5 as the highest expressed FABP family member.

In addition, we investigated the level of expression for the individual family members with qRT-PCR (Figure 1 figure supplement 7C). We examined the relative expression of *FABP3, 4, 5,* and *6* with qPCR in MM.1S, RPMI-8226, and OPM-2 and found that *FABP5* was indeed the predominant family member expressed in our three human MM cell lines and determined that there was no significant effect on the expression of these family members after 24-hour treatment with 50 µM BMS309403, 50 µM SBFI-26, or the combination, suggesting that pharmacological inhibition of FABP5 in MM cells does not induce compensation by other FABP family members, at least at this time point. Future studies should examine the expression of FABP family members in myeloma cells over time and in response to different microenvironments and treatments, since expression of *FABP3, FABP4, FABP5, FABP6* was detected in primary patient samples.

To test the mechanism of action of FABPi, we did the following. We generated Myc inhibitor, (10058-F4), plus FABP inhibitor cell response data in vitro that suggest that FABPs signal through the Myc pathway. We predicted that if Myc is indeed downstream of FABP signaling, as we expect from our RNA-Seq, proteomics, and western blot data, that when the Myc inhibitor (10058-F4) was combined with FABP inhibitors, there would be little to no additive or synergistic effect of the two drugs. However, if the inhibitors were working through different mechanisms, we would observe some synergistic or additive effect. In Figure 4E, we show that BMS309403 alone exerted a consistent and near linear decrease in MM.1S cell number as doses increased, consistent with our prior data demonstrating a 50% decrease in cell number with the 30 µM dose of BMS309403 relative to vehicle control, and an approximate 80% decrease in response to 50 µM. We observed a >50% decrease in cell number when 10058-F4 was added to the cells alone (combined with vehicle), with a minimal effect of the addition of BMS309403 at the majority of doses. Similar effects were observed when SBFI-26 was added to 10058-F4 in Figure 4F. Additional data are provided in Figure 4—figure supplement 1. These data suggest that the effects of the Myc inhibitor are likely acting downstream of the FABP inhibitors.

We additionally explored reactive oxygen species (ROS) signaling. Inhibition of FABPs led to an increase in ROS, including superoxide specifically, (Figure 5A,B; Figure 5 supplement 4-6). We tried to reverse the increase in ROS using an antioxidant, to test the effects of ROS in MM cells, but the antioxidant, N-acteyl cysteine (NAC), was very toxic to the myeloma cells, so we were not able to state that ROS generation was a mechanism of FABP inhibitor effects on MM cells irrefutably.

Reviewer #3 (Recommendations for the authors):1) Despite performing a very comprehensive analysis of the transcriptome, proteome, and metabolic profile of myeloma cells treated with FABPi, the authors did not identify the mechanisms(s) responsible for the decreased cell number. Studies investigating the role of MYC- or UPR-related genes in the response to FAPBi would increase the significance of the findings.

We appreciate the reviewers comment. Since our original submission, we have further tested the role of Myc in the response to FABP inhibition. In an additional cell line, we found that there was a trend towards a decrease in Myc protein level at 48h with combination treatment in 5TGM1-TK cells. In addition, we used a Myc inhibitor, 10058-F4, in combination with various doses of either BMS309-403 or SBFI-26 and found that there was no synergistic effect of the inhibitors, as we described to Review 2 above. Together, our data suggest that inhibition of Myc signaling is one mechanism of action of the FABP inhibitors.

In addition, we investigated the role of reactive oxygen species (ROS) in FABP inhibition in multiple myeloma. Using three cell lines, MM1S, RPMI-8226, and U266, we investigated total ROS with CellROX, and superoxide, with MitoSOX over 72 hours of FABP inhibitor treatment. We were additionally able to rescue the induced ROS effect with N-acetyl cysteine, an antioxidant, but this did not inhibit apoptosis from occurring with FABP inhibitor treatment, as we described above to Reviewer 2.